# A task-invariant prior explains trial-by-trial active avoidance behaviour across gain and loss tasks
Tobias Granwald [1,2] ✉, Peter Dayan [3,4], Máté Lengyel [5,6] & Marc Guitart-Masip [1,2,7] ✉

Failing to make decisions that would actively avoid negative outcomes is central to helplessness. In a Bayesian framework, deciding whether to act is informed by beliefs about the world that can be characterised as priors. However, these priors have not been previously quantified. Here we administered two tasks in which 279 participants decided whether to attempt active avoidance actions. In both tasks, participants decided between a passive option that would for sure result in a negative outcome of varying size, and a costly active option that allowed them a probability of avoiding the negative outcome. The tasks differed in framing and valence, allowing us to test whether the prior generating biases in behaviour is problem-specific or task-independent and general. We performed extensive comparisons of models offering different structural explanations of the data, finding that a Bayesian model with a task-invariant prior for active avoidance provided the best fit to participants' trial-by-trial behaviour. The parameters of this prior were reliable, and participants' self-rated positive affect was weakly correlated with this prior such that participants with an optimistic prior reported higher levels of positive affect. These results show that individual differences in prior beliefs can explain decisions to engage in active avoidance of negative outcomes, providing evidence for a Bayesian conceptualization of helplessness.

Getting out of bed this morning was a choice to be active. It may even have been a difficult one if you are suffering from depression[1]. Initially, staying in bed might sound attractive, but if you did, you would incur a range of negative consequences. However, getting up also has real costs, and even if you do so, the negative consequences that you tried to avoid may still occur. These types of decisions of active avoidance in which a choice not to do something—to be passive – results in a negative consequence, while an active action may, or may not, result in escaping that consequence, are central to conceptualisations of helplessness[2–5]. As such, characterising how people engage in these decisions is key to understanding helplessness, and in the long run, depression.

When deciding whether to get up or not, your beliefs about how much your actions can affect your outcomes have a key impact. According to Bayesian theories of decision-making[6–10], such beliefs are determined by two quantities: the initial prior, and previous experience with the consequences of the actions, formalised as the likelihood. The updating of prior beliefs based on this experience can then be computed as the Bayesian combination of

prior and likelihood, giving rise to the posterior. The prior plays a particularly critical role in beliefs about action outcomes as the decision-maker will only opt for active choices and thus obtain experience about their outcomes at all if the initial prior is adequately optimistic in the first place[11,12].

Recent research has pointed to the importance of prior beliefs in depression and helplessness. For example, prior beliefs about the outcomes of actions in a visuomotor task may be correlated with measures of behavioural activation[13], a concept closely related to anhedonia and helplessness[14]. Another study showed that more pessimistic prior beliefs about one's own scholastic ability were specifically linked to depressive symptoms[15], further indicating how priors may be altered in depression. However, these studies only employed single behavioural tasks, and so they were not suitable to test the existence of the sorts of priors typically assumed in theoretical work that generalise broadly[2,3,16–18]. The current study aims to characterise how individual differences in prior beliefs may make people less or more prone to choose to be passive across different contexts.

[1]Aging Research Center, Department of Neurobiology, Care Sciences and Society, Karolinska Institutet and Stockholm University, Stockholm, Sweden. [2]Center for Cognitive and Computational Neuropsychiatry (CCNP), Karolinska Institutet, Stockholm, Sweden. [3]MPI for Biological Cybernetics, Tübingen, Germany. [4]University of Tübingen, Tübingen, Germany. [5]Computational and Biological Learning Lab, Department of Engineering, University of Cambridge, Cambridge, UK. [6]Center for Cognitive Computation, Department of Cognitive Science, Central European University, Budapest, Hungary. [7]Center for Psychiatry Research, Region Stockholm, Stockholm, Sweden. ✉e-mail: tobias.granwald@ki.se; marc.guitart-masip@ki.se

Learned helplessness has been a particular target for such Bayesian analyses. In this phenomenon, a person or an animal first experiences uncontrollable aversive outcomes in one setting and later does not attempt to avoid aversive outcomes in another setting or context in which those outcomes could actually be avoided[4,5]. This paradigm of reduced active avoidance has long been utilised as an animal model of depression[19]. An important notion for learned helplessness is that of the controllability of the environment, i.e., the extent to which an agent believes that their actions have a causal influence over the outcomes they receive[2,3,20,21]. In a Bayesian framework, learned helplessness has been conceptualised as resulting from a pessimistic prior (acquired from initial outcomes) that generalises across different contexts and implies the expectation of failure when performing an action[2,3]. Here we seek to test the hypothesis that helpless behaviour is underpinned by such a pessimistic prior by estimating it from trial-by-trial behaviour independently across different tasks.

The current study builds upon the methods used to extract generalised priors in other cognitive domains. In a key study, Houlsby and colleagues[16] developed a method to extract complex multidimensional priors that generalise across tasks. They used this method in two different face-perception tasks to extract a task-invariant prior for this domain[16]. Building on this method, in the current study, we employed two different tasks with the aim of quantifying priors and testing whether they generalise. In the tasks, participants made repeated binary decisions between a sure negative consequence and a costly active choice affording a non-zero probability of avoiding the negative consequence. This choice structure is similar to that outlined in simulations of learned helplessness[3]. The tasks were structured so as to satisfy two requirements: having sufficient trials in which an initial prior established outside the experimental condition will exert substantial influence on participant's behaviour, and revealing how this prior is updated based on the outcomes they observe. To test generalisability, the two tasks also differed in several important ways, including their framing, the surface features of stimuli, and valance of the different potential outcomes. This structure allowed us to test whether participants construct a common generalised prior for action outcome contingencies, as assumed by simulations of learned helplessness[3], or if these types of priors are specific to each task.

To test for convergent validity, we collected data on participants' self-rated positive affect, as assessed by the positive and negative affect schedule[22] and trait anxiety, as assessed by the state-trait anxiety index—trait (STAI-T)[23]. Reduced positive affect has been specifically linked to depression[24,25], thereby providing a measure with high variability in a general population sample that is related to depression[26]. Trait anxiety scores correlate with depression in non-clinical samples[27] and more generally correlate both with depression and anxiety[28]. Furthermore, based on our pilot data (see Supplementary Note 2, pilot data), we have previously observed a correlation between the relative variance of the task-invariant prior and trait anxiety.

We hypothesised and preregistered that participants' behaviours would be affected by the values associated with the options and whether previous active choices had been successful in avoiding negative outcomes. We further hypothesised that their behaviour in both tasks would be best described by a Bayesian choice model with a shared prior across the tasks. Furthermore, we also hypothesised that this prior would be reliable within the session and two weeks later and correlate with participants' self-rated positive affect and trait anxiety.

## Methods
The research plan for this study was preregistered before the start of the data collection on 2024-01-09 at OSF: https://osf.io/rej74.

### Participants
Three hundred sixty-eight participants between 18 and 40 years old were recruited in a sex stratified manner, between 2024-01-22 and 2024-01-26, from the online participant pool Prolific (www.prolific.co). Three hundred two of these participants completed the experiment. Out of these participants, 21 participants' data were lost due to technical issues, giving a sample of 281 cases of complete data. One participant was rejected due to failing more than two out of the eight (two in each block) clearly telegraphed attention checks, where participants were asked at two times in each block to press the arrow key up before a timer of 10 s ran out. Four of these attention checks (one in each block) were shown directly following the last instructions before the start of the block, and four (one in each block) were shown at randomly selected trials after the last trial with a stimulus. One participant was removed due to failing to make a response in more than ten trials, thereby repeating these trials. This provided a final sample size of 279 (133 self-reported female participants, 142 self-reported male participants; four participants without demographic data). The participant age ranged from 18 to 40 ($M = 27.12$, SD = 5.06), with participants from 36 different nationalities (see Supplementary Methods for a full breakdown of sample characteristics).

Because the participants who encountered technical issues were still compensated for their participation, budgetary constraints precluded us from reaching our preregistered target sample size of 300 participants. This target was based on an a priori power simulation where 1000 simulated model comparisons were carried out with parameter values and model frequencies corresponding to the second block of a pilot study with 54 participants. Simulations showed that at a sample size of 300 subjects, there is a 91.8% probability of observing a protected-exceedance probability above 0.95 in favour of the 1-prior Bayesian learner model. 300 subjects also provide a 99.9 and 80.6% probability of observing a protected-exceedance probability above 0.95 in favour of the Bayesian learner model for the robber and factory task, respectively. This sample size also provides 90.6% power for the correlation between the mean of the prior and PANAS-P based on the observed effect size ($r = 0.169$) in the pilot study with an alpha-level set at 0.05. The sample size further provides 99.9% power for the correlation between relative variance of the prior and STAI-T based on the observed effect size ($r = 0.328$). As such, while we did not reach our target sample size, the current sample is appropriately powered to test each hypothesis.

Participants were paid 10 £ if they agreed to participate and completed the experiment on the first day. Participants had a chance to earn up to an additional 4 £ if they gained more points in the tasks than the mean of all participants in a randomly selected trial. A random subset of 101 participants was then invited back to participate in the follow-up experiment about two weeks later (the time between sessions was $M = 16.02$ days, SD = 3.09 days, range = 12 to 27 days). The time between sessions differed from the preregistered 1-week follow-up because of limitations in the recruitment process, whereby we were unable to specify a follow-up time. In this session, participants were paid a further 5 £ if they agreed to participate and completed the experiment, as well as another 2 £ based on their performance, as on the first day. Before participating both on the first day and the follow-up, participants were required to provide informed consent for their participation in the study. The study complies with all the relevant ethical regulations for research with human participants. The study protocol was approved by the Swedish Ethical Review Authority (ref.no.: 2021-04906) before the start of the study.

### Design
The study consisted of a within-subjects design in which each participant performed the two decision-making tasks (see detailed descriptions of each task below). Participants performed the tasks online, through their web browser. The experiment was hosted on a server at pavlovia.org. The experiment was programmed in JavaScript, HTML and CSS using the experiment toolbox jsPsych v.7.1.2[29].

Participants performed the tasks multiple times with different stimuli. After providing informed consent, participants performed two blocks of both tasks on the first day (($60 + 60$) + ($60 + 60$) trials). The task that participants began each day was randomly selected. After that, the tasks strictly alternated between blocks. Before starting the first block of each task, participants performed 18 practice trials with four different stimuli and three different probabilities of success when choosing the active choice (number of trials and success probability with each stimulus in a

randomised order = 4 trials with 80%, four trials with 20% and four and six trials with 50%) to familiarise them with the structure of the task. The stimuli in the practice trials were not repeated in the real task. After both blocks of the decision tasks were completed, participants ended the first day by completing a selection of questionnaires (see below and in Supplementary Methods, Questionnaires) and a short memory task[30] that was unrelated to the hypotheses of the current study. After completing this part of the study, a subset of random participants was invited back for a follow-up session. This time, participants performed one block of each task (60 + 60 trials). As on the first occasion, participants completed practice trials before each block. Participants also completed the same questionnaires after the tasks as on the first day. Data collection and analysis were not performed blind to the conditions of the experiments.

**Robber task**. This task was described as an encounter with a robber. Participants were asked to imagine being a server at a bar. They were told that at the end of the night, they were on their way home with their tips from the night in hand (the offer) when they encountered a robber. Participants could choose to relinquish their money (passive) or try to fight the robber (active). If they handed over the money, they would be missing out on their potential reward. If they tried to fight the robber, they incurred the cost of fighting. If successful, they kept the offer minus the cost, but if unsuccessful, they missed out on the reward associated with the offer and incurred an additional loss amounting to the cost of fighting. On each trial, $t$, the offer ($o$) was sampled from the 20-element array presented in Eq. (1):

$$o_t \in \frac{50}{[0.10, 0.15, \ldots, 0.95, 1, 2]} \tag{1}$$

and then rounded to the nearest whole number. Each offer was presented 3 times within a block in a randomised order. In this task, the action cost was described as the cost of fighting. Crucially, this cost was related to the size of the tip by constructing the cost as a sum of a percentage of the offer and a set cost. In the robber task, the set costs were 10, 40, or 75 points, and the concomitant percentages of the offers were 15, 5 and 0%. This resulted in 60 unique offer-cost combinations, with the majority of these combinations resulting in small differences between the values of the active and passive options two options (see Fig. 1C). The order of the combination of offers and costs was shuffled across trials. Participants were told that they were supposed to try to get as many points as possible and that one of their trials would be randomly selected, and they would gain a bonus based on their performance on that trial.

Within each block of the task, participants encountered 12 different robbers (36 unique stimuli in total across the first day and follow-up). Each robber was signalled with a cartoon drawing of a robber. Each of these stimuli was associated with a unique probability of successfully winning the fight if the active choice was chosen. The success probabilities ranged from 7.69 to 92.31% in equidistant steps (for the full range of success probabilities and some of the robbers see Fig. 1G). This setup resulted in each stimulus signalling a unique success probability. Crucially, this success probability was not known by the participants. Participants encountered each stimulus in four to eight consecutive trials. Six of the stimuli were encountered four times, three stimuli five times and one stimulus six, seven and eight times. The number of encounters for each stimulus was balanced over the block by splitting the number of encounters into two counterbalanced lists and shuffling the number of encounters within the lists. This ensured a spread of few and many encounters with the stimuli across the task. This design allowed us to capture Bayesian updating, as it encouraged some learning between repetitions of the same stimulus without making the task too simple.

**Factory task**. This task was described in terms of decisions related to maintenance work. Participants were asked to imagine being the foreperson of a factory in which a machine was breaking down. Participants decided whether to let the machine break down (passive) or to try to repair the machine (active). The costs within the factory task were described as the repair cost of the machine, while the offer was the money they lost if the machine broke down. This setup means that, unlike the robber task, the values in the factory task were exclusively negative. If they let the machine break down, participants incurred a sure loss. If they tried to repair the machine and were successful, they only incurred the cost of repairing, but if unsuccessful, they lost the value of the machine and the cost of repairing. To make the ranges of values comparable across the tasks, the offers in the factory task were the same as the robber task, but in the negative domain. However, to distinguish the tasks in values of the actions while keeping the same range of values, the factory task had a different set of costs associated with the action. Much like the robber task, the action cost shown to the participants was constructed with a set cost and a cost associated with a percentage of the offer. The corresponding set and percentage costs for the factory task were 0, 50 or 100 points and 10, 5 and 0%, also resulting in small differences between the two options for the majority of combinations (see Fig. 1D). Participants were told that they were supposed to try to lose as few points as possible and that one of their trials would be randomly selected and they would gain a bonus based on their performance on this trial.

As in the robber task, participants encountered 12 different types of machines in each block (36 unique stimuli across the first day and follow-up). Each machine was signalled with a different drawing of a machine. The drawings were bought from a website for royalty-free vector images (https://www.vectorstock.com/) and were chosen from four different sets of machine drawings drawn by four different artists (vikivektor, WinWin_artlab, Tarlia and MicroOne). The images feature different factory machines and were chosen to have a similar style of drawing but with distinct features (see Fig. 1G). Like the robbers of the robber task, each machine type signalled a unique probability of being successful in repairing the machine, with the range of success probabilities being the same across tasks. Likewise, the number of trials in which each stimulus was encountered mirrors those in the robber task. However, the order of the trials and success probabilities were randomised separately for each task.

## Questionnaires
To establish convergent validity, participants completed two questionnaires, the Positive and Negative Affect Schedule (PANAS)[22] and the State-Trait Anxiety Index (STAI)[23]. Beyond these two preregistered questionnaires, we collected data from a range of questionnaires which is not part of this study. These questionnaires are presented in the Supplementary material (Supplementary Methods, Questionnaires). The order in which the questionnaires were presented was randomised for each participant.

**Positive and negative affect schedule (PANAS)**. To measure participants' degree of positive affect, we used the Positive and Negative Affect Schedule (PANAS)[22]. The PANAS is a 20-item scale consisting of two subscales measuring positive affect and negative affect, respectively. The items consist of words that describe different feelings or emotions, and respondents are asked to rate to what extent they have felt the specific feeling over a specific time period. For the current study, we chose over the past week. Participants rated the items on a five-point scale ranging from 1 "*very slightly or not at all*" to 5 "*extremely*". For the focus of this study, we only considered the positive affect subscale, but both scales were administered. The positive affect scale has good internal consistency in non-clinical samples ($\alpha = 0.89$[26]). In the current study, we find good internal consistency ($\alpha = 0.912$) and moderate test-retest reliability ($ICC(2,1) = 0.733$, 95% CI = [0.628, 0.811]). Low positive affect, as measured with the positive affect subscale, correlates with symptoms of depression in clinical[24,31] and nonclinical[26] samples, in particular symptoms of hopelessness, anhedonia, lack of energy and worthlessness[24]. Furthermore, the positive affect scale, as compared with the negative affect scale, has been found to be more strongly correlated with symptoms of depression[24,26,31]. Furthermore, the positive affect subscale has also been shown to be more strongly correlated with symptoms of depression

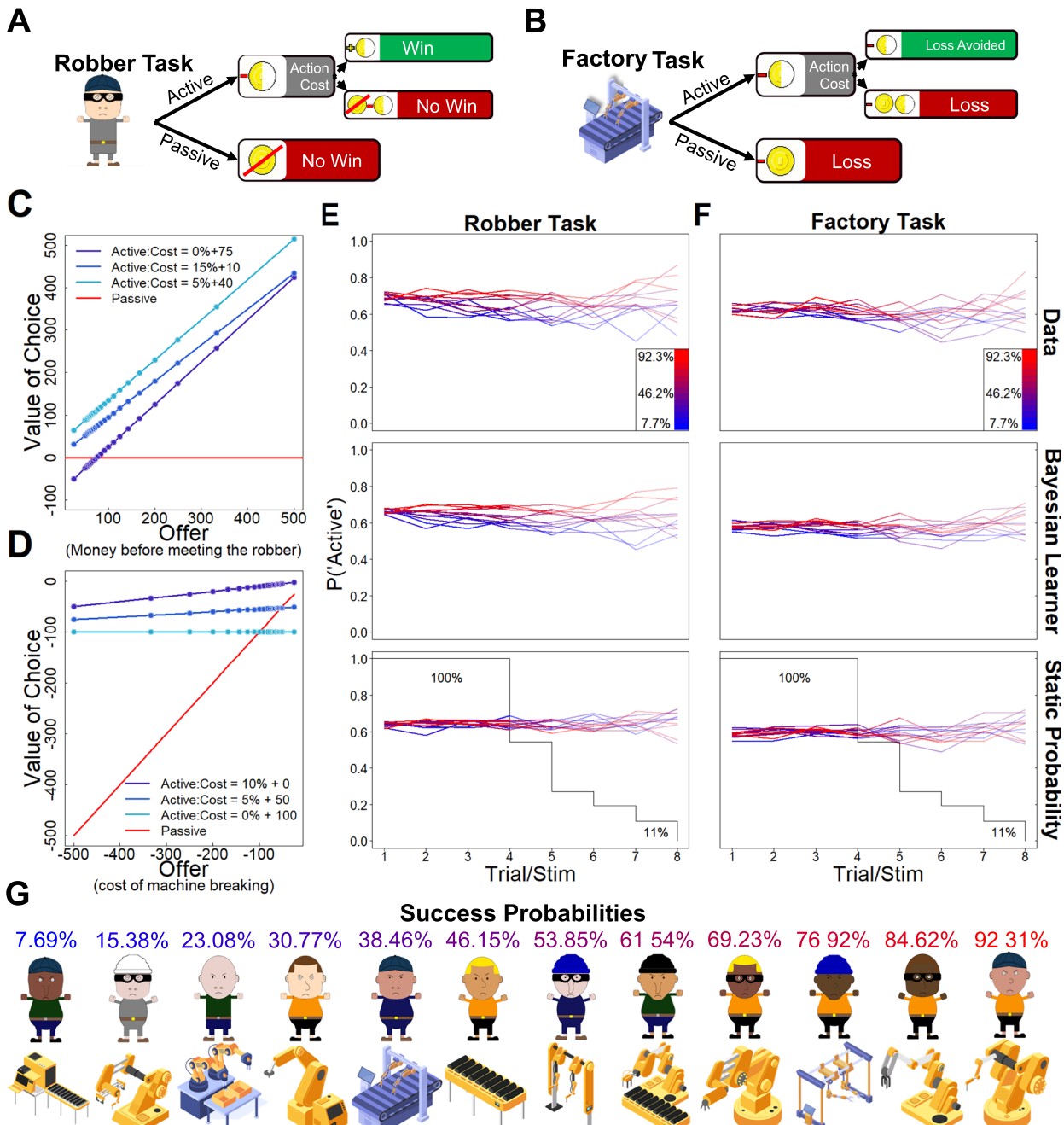

**Fig. 1 | Values and actions in the experiment. A** Schematic layout of the decision structure in the robber task. **B** Schematic layout of the decision structure in the factory task. **C** Offers and costs in the tasks. The x-axis shows the offers for each task. In the robber task, the offer is the amount of the tip that the participants may earn at the end of the night before encountering the robber. The three blue lines show the values of the active choice (if successful) at the different cost regimes employed in the tasks. The red lines show the value of the passive choice. The dots show the specific values present in the task. **D** Same as D but for the factory task. In the factory task, the offers are the possible losses that the participant incurs if they let the machine break. **E** Decisions over trials in the first block of the robber task. The mean probability of choosing the active option over trials with the same stimulus is shown for different success probabilities (colours in the legend in the top plots and G), in data (top; $N = 279$) and in simulated behaviour (middle and bottom). The line shown in the bottom two plots and the opacity of the lines indicate the percentage of participants that encountered each success probability on each trial, as the number of times each participant encountered each of the success probabilities ranged from 4 (100%) to 8

(11%) trials. The lines thereby signal the increased uncertainty in the estimates, since not all participants encountered all stimuli eight times. The middle and bottom plots show mean model behaviour across 1000 simulations, simulated with the participants' parameter values and design matrix (order of success probability, offers, costs and trials per stimuli) with the Bayesian learner (middle) and Static probability model (bottom) fitted to the first block of the robber task. **F** Same as E for the first block of the factory task ($N = 279$) and replacing the basic Static probability model with the Static probability model with bias ($\varphi$) parameter. **G** Example stimuli that the participants could encounter in the first block of either task, along with the success probabilities that were assigned to each stimulus. The order of the stimulus presentation and the association of stimuli to success probabilities was randomised for each participant. Participants encountered each stimulus in four to eight consecutive trials. Six of the stimuli were encountered four times, three stimuli five times, and one stimulus six, seven and eight times. The number of encounters for each stimulus was balanced over the block by splitting the number of encounters into two counterbalanced lists and shuffling the number of encounters within the lists.

**Table 1 | Models tested when fitting the tasks separately**

| Name | N | Utility function | | Choice function |
|---|---|---|---|---|
| SoftMax Only | 1 | Robber : | $\begin{cases} Q_t(\text{go}) = o_t - c_t \\ Q_t(\text{ng}) = 0 \end{cases}$ | |
| | | Factory : | $\begin{cases} Q_t(\text{go}) = -c_t \\ Q_t(\text{ng}) = -o_t \end{cases}$ | |
| Win-Stay Lose-Shift | 3 | Robber : | $\begin{cases} Q_t(\text{go}) = (o_t - c_t) + (\text{success}_{t-1} \times \gamma^+) - (\text{loss}_{t-1} \times \gamma^-) \\ Q_t(\text{ng}) = 0 \end{cases}$ | |
| | | Factory : | $\begin{cases} Q_t(\text{go}) = -c_t + (\text{success}_{t-1} \times \gamma^+) - (\text{loss}_{t-1} \times \gamma^-) \\ Q_t(\text{ng}) = -o_t \end{cases}$ | $P_t(\text{go}) = \frac{1}{1+e^{-(Q_t(\text{go})-Q_t(\text{ng}))/\tau}}$ |
| Static Probability | 2 | Robber : | $\begin{cases} Q_t(\text{go}) = -c_t + p \times o_t \\ Q_t(\text{ng}) = 0 \end{cases}$ | |
| | | Factory : | $\begin{cases} Q_t(\text{go}) = -c_t - (1-p) \times o_t \\ Q_t(\text{ng}) = -o_t \end{cases}$ | |
| Bayesian Learner | 3 | Robber : | $\begin{cases} Q_{t,j}(\text{go}) = -c_{t,j} + \mu_{0+t,j} \times o_{t,j} \\ Q_{t,j}(\text{ng}) = 0 \end{cases}$ | |
| | | Factory : | $\begin{cases} Q_{t,j}(\text{go}) = -c_{t,j} - (1 - \mu_{0+t,j}) \times o_{t,j} \\ Q_{t,j}(\text{ng}) = -o_{t,j} \end{cases}$ | |

The table shows the names, number of parameters (N) and utility and choice functions of these models.

than symptoms of anxiety[24,25,32]. In the current study, we used the summed score of the full PANAS-P subscale.

**State-trait anxiety index (STAI).** To measure the participants' degree of nonspecific negative affectivity, we used the State-Trait Anxiety Index (STAI-Y-form[23]). The STAI-Y is a 40-item scale consisting of two subscales, each comprised of 20 items, and is the most cited trait anxiety scale[28]. The state subscale measures how anxious a person is right now, at this moment. The trait subscale measures how anxious a person generally feels. For the focus of this study, we only considered the trait subscale STAI-T (Y-2), but both scales were administered. Participants rated the items on a four-point scale ranging from 1 (almost never) to four (almost always), with nine reverse-coded items. Despite the initial aim of the scale being to measure anxiety, a recent meta-analysis has shown that, across studies, individuals with diagnosed depression show higher STAI-T scores than persons with anxiety disorders[28]. As such, we will follow these authors' proposal that the scores should be interpreted as nonspecific negative affect[28]. Considering this, the scale has good convergent validity with both scales measuring depression (mean $r = 0.61$) and anxiety (mean $r = 0.59$)[28]. The STAI-T has also shown good test-retest reliability (mean $r = 0.88$[33]) and internal consistency (mean Cronbach's alpha = 0.89[33]). In the current study, we find good internal consistency ($\alpha = 0.916$) and moderate test-retest reliability (ICC(2,1) = 0.857, 95% CI = [0.795, 0.901]). In the current study, we used the summed score of the full STAI-T subscale with the appropriate items reverse-coded.

**Analysis**

To test the study's hypotheses, we used computational models and model comparison, Bayesian hierarchical regressions and Spearman's correlations. In the sections below, we will first introduce the computational models that we used, then we will show how these, and other methods, were used to test our hypotheses. Lastly, we will present our inference criteria.

**Method of parameter estimation and model comparison.** Parameters in all computational models were estimated with the hierarchical Bayesian inference (HBI) algorithm that is part of the computational and behavioural modelling (CBM) toolbox v. from 04/2019[34] in MATLAB v. 9.13.0.2105380 (R2022). This method was also used for model comparison. In HBI, parameter estimation is carried out in two steps, first by performing an initial Laplace approximation of the parameters at the individual level using a predefined non-informative prior, followed by iteratively regularising and re-estimating the parameter across the full

group of individuals taking into account how likely each individual is to be best fitted by a specific model. It further represents all uncertainties at both individual and group levels as probability distributions. Taken together, these steps result in HBI having smaller errors in estimation compared with other Bayesian hierarchical and non-hierarchical inference methods[34]. In model comparison, HBI applies a form of what is known as an automatic Occam factor in which more complex models are directly penalised by spreading probability mass more broadly, thereby balancing model complexity and goodness-of-fit. This stems from the extra level of the Bayesian hierarchy that it includes compared with the type-2 maximum likelihood iBIC method employed by Huys et al.[35] Each parameter in all models had a prior that was a zero-centred normal distribution with a variance of 6.25 as recommended by the authors of the toolbox[34]. The parameters were transformed to their respective parameterisations within the models. Parameters with support ranging from 0 to $+\infty$ were exponentiated (implying a log-normal prior) while parameters with support restricted from 0 to 1 were transformed with a standard logistic transformation.

**Cognitive models.** A full description of all models is given here.

The parameters of four different types of choice models were estimated when fitting the tasks and blocks separately (see Table 1). The four models are: a SoftMax-only model (one parameter), a win-stay lose-shift model (three parameters), a Static probability model (two parameters) and a Bayesian learner model with a beta distributed prior (three parameters). The SoftMax-only model has only one parameter, the temperature ($\tau$), and employs a utility function that assumes a 100% probability of success if the active choice (go) is taken. The support of the $\tau$ parameter in this and all other models is 0 to $+\infty$, so we reparametrize it with its logarithm (which is unbounded). This parameter was also multiplied by 0.01 to improve sampling of smaller values.

Equation (2) shows the SoftMax choice function quantifying the probability of choosing the active option on trial $t$ ($P_t(\text{go})$) based on the expected values ($Q$) of the active (go) and passive (ng) options: this choice function is the same across all four models, and Eq. (3) shows the utility functions in the robber and factory tasks in the SoftMax-only model, which implicitly assumes a 100% probability of success for active choices. As such, this model simply uses the offers ($o$) and costs ($c$) on each trial ($t$) to calculate the expected value of each option.

$$P_t(\text{go}) = \frac{1}{1 + e^{-(Q_t(\text{go}) - Q_t(\text{ng}))/\tau}} \quad (2)$$

$$\text{Robber} : \begin{cases} Q_t(\text{go}) = o_t - c_t \\ Q_t(\text{ng}) = 0 \end{cases}$$
$$\text{Factory} : \begin{cases} Q_t(\text{go}) = -c_t \\ Q_t(\text{ng}) = -o_t \end{cases} \tag{3}$$

Beside using Eq. (2) as a choice function (thus including the $\tau$ parameter) and assuming a 100% probability of success, the Win-stay lose-shift model includes two more parameters. These provide a positive ($\gamma^+$) or negative bonus ($\gamma^-$) to the utility of the active choice, depending on the outcome of the previous trial if an active choice was taken. This implemented a simplified, stimulus-independent, heuristic learning rule[36]. The support of these parameters is also 0 to $+\infty$, and so they are also reparametrized with their logarithms. Equation (4) shows the utility functions of the Win-stay lose-shift model.

$$\text{Robber} : \begin{cases} Q_t(\text{go}) = (o_t - c_t) + (\text{success}_{t-1} \times \gamma^+) - (\text{loss}_{t-1} \times \gamma^-) \\ Q_t(\text{ng}) = 0 \end{cases}$$
$$\text{Factory} : \begin{cases} Q_t(\text{go}) = -c_t + (\text{success}_{t-1} \times \gamma^+) - (\text{loss}_{t-1} \times \gamma^-) \\ Q_t(\text{ng}) = -o_t \end{cases} \tag{4}$$

Here, $\text{success}_{t-1} = 1$ if the last trial's choice was an active choice and the participant was successful and 0 if the participant chose the passive choice or lost. Similarly, $\text{loss}_{t-1}$ is 1 if the last choice was an active choice and the participant was unsuccessful and 0 if they chose the passive choice or were successful in their active choice.

The Static probability model also uses Eq. (2) as a choice function (and thus includes the $\tau$ parameter). However, in contrast to the two previous models, this model does not assume a 100% probability of success. Instead, the model includes a free parameter for the participant's subjective probability ($p$) of success, which is assumed to apply across all robbers and machines. This parameter's support is 0 to 1, and it is reparametrized with its logit. Equation (5) shows the utility functions for this model.

$$\text{Robber} : \begin{cases} Q_t(\text{go}) = -c_t + p \times o_t \\ Q_t(\text{ng}) = 0 \end{cases}$$
$$\text{Factory} : \begin{cases} Q_t(\text{go}) = -c_t - (1-p) \times o_t \\ Q_t(\text{ng}) = -o_t \end{cases} \tag{5}$$

The Bayesian learner model uses the same utility functions, but instead of having a single set value for the subjective probability of success, this model assumes Bayesian updating of stimulus (i.e., robber/machine)-specific success probabilities. According to this model, participants start with a beta distributed prior over success probabilities before encountering a stimulus, and then compute a new posterior over subjective probabilities of success after each active choice. These posteriors (and, on the first occasion, the initial prior) are the predictive distributions this model uses to compute the probability of success on any given trial.

Specifically, the parameters estimated in this model are the mean ($\mu_0$) and relative variance ($\sigma_0^2$) of the initial prior. The relative variance is simply the variance of the beta distribution divided by the upper bound of the variance for a given value of $\mu_0$ (which is $\mu_0 \cdot (1-\mu_0)$). As such, the support for both $\mu_0$ and $\sigma_0^2$ is 0 to 1, and they are thus reparametrized with their logits. These parameters are then transformed into the $\alpha$ and $\beta$ parameters of the beta distribution in Eq. (6).

$$\text{prior} : \begin{cases} \alpha_0 = \mu_0\left(\frac{1}{\sigma_0^2} - 1\right) \\ \beta_0 = (1 - \mu_0)\left(\frac{1}{\sigma_0^2} - 1\right) \end{cases} \tag{6}$$

This transformation is used to simplify the Bayesian updating of this prior based on the outcomes of the active choices. In particular, due to the conjugacy of a beta prior with Bernoulli (binary) observations, the Bayesian posterior for the success probability of stimulus $j$ after trial $t$ is also a beta distribution with alpha and beta parameters simply given as the sum of the number of successes and losses incurred up until trial $t$ and the alpha and beta parameters of the initial beta prior distribution, respectively. Thus, the parameters of the posterior for stimulus $j$ up until trial $t$ are computed recursively as in Eq. (7).

$$\alpha_{t,j} = \begin{cases} \alpha_{t-1,j} + 1 & \text{success}_{t-1,j} = 1 \\ \alpha_{t-1,j} & \text{otherwise} \end{cases}$$
$$\beta_{t,j} = \begin{cases} \beta_{t-1,j} + 1 & \text{loss}_{t-1,j} = 1 \\ \beta_{t-1,j} & \text{otherwise} \end{cases} \tag{7}$$

Note that after an active choice, either alpha or beta is incremented, depending on success, while after choosing to be passive, both parameters remain unchanged. The mean of the posterior is then used in the utility function as the subjective probability (Eq. (8)).

$$p_{t,j} = \frac{\alpha_{t,j}}{\alpha_{t,j} + \beta_{t,j}} \tag{8}$$

Note that on the first trial of each stimulus, this is the same as $\mu_0$ (the free parameter we estimate). The inclusion of stimulus-specificity in this model is a consequence of the fact that the model assumes that participants revert to their prior with each new stimulus.

We also fitted two augmented versions of the Bayesian model. These augmentations are tested as an attempt to remove noise from the estimated parameters of interest and to test alternative hypotheses about the extent of learning. One augmentation was a model that did not revert to the prior with each new stimulus but instead continued to update the posterior from the last trial regardless of stimulus identity. This challenges the assumption that updates only occur within each stimulus. In the second augmentation, we assumed that participants might have an action bias either towards action or inaction (passivity). This is implemented as a lapse parameter by adding a parameter ($\varphi$) to the choice function as follows:

$$P_t(\text{go}) = [\varphi]_+ + (1 - |\varphi|)\frac{1}{1 + e^{-(Q_t(\text{go}) - Q_t(\text{ng}))/\tau}} \tag{9}$$

Where $[\varphi]_+$ is a rectified linear function where:

$$[\varphi]_+ = \begin{cases} \varphi & \varphi \geq 0 \\ 0 & \text{otherwise} \end{cases} \tag{10}$$

And $|\varphi|$ is the absolute value function. As such, the support for $\varphi$ in this model is $-1$ to 1. Values of $\varphi < 0$ implement a passivity bias, whereas $\varphi > 0$ implements an action bias. This implementation thereby assumes that there is a probability that the participant completely neglects the value calculation in preference for choosing a specific option. When surviving model comparison, this prevents decision noise caused by these types of decisions from affecting the estimate of the prior or other model parameters, much in the same way as implementations of lapse parameters in psychometric functions can improve the fit of these functions[37]. To ensure that we could estimate the likelihood as a function of $\varphi$ using gradient-based optimisation, we replaced the rectified linear (whose gradient is undefined at $\varphi = 0$) with a SoftPlus function with the parameter alpha set at 100. This method is described in detail in the Supplementary Methods.

When fitting the tasks together, we utilised two different versions of the best-fitting version of the models for each task separately. One version assumed different priors ($\mu_0$ and $\sigma_0^2$) for the different tasks, and the other assumed a single, shared prior across tasks. Both models allowed different temperature parameters ($\tau^R$ and $\tau^F$) across tasks. This ensures that task-specific noise does not affect the estimate of the parameters.

Model recovery was performed for each model comparison (see Supplementary Figs. S9–S13) and parameter recovery was performed for the best-fitting models (see Supplementary Tables S22, S23 and Supplementary

https://doi.org/10.1038/s44271-025-00254-1                                                                                **Article**

Figs. S14, S15). See Supplementary Note 1: Supplementary Results, for a detailed description of these analyses.

**Statistical analyses.** We performed both model-agnostic and model-based analyses of the participants' choice data.

To perform a model-agnostic analysis of participants' choice behaviour, we used Bayesian hierarchical logistic regression. These models were used to analyse whether participants' choice behaviour is affected by the outcomes of the previous trial with the same stimulus and the offers and costs of a specific trial. Both models were fit using Markov chain Monte Carlo (MCMC). Models were fit using the package *brms* v.2.19.0[38,39] in R v. 4.3.0 and Stan. The models were fitted with weakly regularising Gaussian priors with $M = 0$ and varying standard deviations for the intercept (SD = 1.5), population (SD = 1) and group-level effects (SD = 1). The prior for the group-level effects correlation matrix was set with a Lewandowski–Kurowicka–Joe (LKJ) distribution with the parameter set at 3 thereby favouring somewhat weaker correlations between group-level effects[40,41]. The models were sampled with 4500 iterations in seven chains, with 1000 samples per chain discarded as a warmup. The sampling was done with the default treedepth of 10 and a delta of 0.8 in all models except when there were more than 50 divergent transitions after warmup. Delta was altered for model 2 when fitted in both blocks of the factory task (here delta = 0.95) and in the second block of the robber task (delta = 0.98). All models converged with Rhat = 1.00 for the population-level effects (for full Rhat and trace plots of the sampling see Supplementary Figs. S1–S8).

We fitted two different Bayesian hierarchical logistic regressions, specified as stated below in the tasks and blocks separately. Regression model 1 was specified as follows:

$$\text{Choice} \sim \text{Intercept} + (o_t - c_t) \times \text{success}_{t-1}$$
$$+ (\text{Intercept} + (o_t - c_t) \times \text{success}_{t-1}|\text{subject})$$
$$+ (\text{Intercept} + (o_t - c_t) \times \text{success}_{t-1}|\text{stimulus})$$

Regression model 2 was specified as follows:

$$\text{Choice} \sim \text{Intercept} + \text{trial number with same stimuli}$$
$$\times \text{Success probability of the stimuli}$$
$$+ (\text{Intercept} + \text{trial number with same stimuli}$$
$$\times \text{Success probability of the stimuli}|\text{subject})$$

When correlating parameters across tasks, we used Spearman's ($r_s$) correlations and performed a permutation test with 10,000 random permutations to test the true correlation. We also reported the parameters' explained variance ($R^2$) from one task to the other. In a follow-up post-hoc analysis, which was not part of the preregistered protocol, we also explored potential differences between the tasks in the percentage of active choices and successful active choices. Here, we used Wilcoxon signed-rank tests.

The reliability of the parameters was tested with a two-way mixed effects, absolute agreement, single measure, interclass correlations (ICC(2,1)). To perform this analysis, we used the R package *irr* v.0.84.1[42]. Because of the assumption of normality for this test, the ICC was carried out on the untransformed parameters of the model, which were estimated with a normally distributed prior, and as such, they were assumed to be normally distributed (although this was not formally tested).

To test the convergent validity of the priors extracted the parameter values from the best-fitting model. Participants' parameter values were then correlated with each participant's sum score in positive affect and STAI-T using Spearman's correlations.

**Inference criteria.** For frequentist statistical analysis, the α-level was set at 0.05 meaning that *p* values below this threshold were considered statistically significant. For Bayesian analysis, the Bayes factor (*BF*) was used to determine evidence in favour of or against a hypothesis. A Bayes factor of $BF_{10} > 10$ was considered enough evidence to reject the null hypothesis and a $BF_{10} < 0.1$ was considered enough evidence to reject the alternative

hypothesis, values in between were considered inconclusive. All analyses were tested with two-sided hypothesis testing. For computational model comparison, the best-fitting model at the group level was selected based on protected-exceedance probability (*PXP*)[43]. *PXP* provides a continuous measure of the probability that a specific model is the most prevalent in the population. A *PXP* above 0.95 was considered strong evidence for a given model. However, values bigger than chance (1/number of models concerned) were also considered evidence supporting the model. For each model comparison, we also reported the model frequency (*M.freq*) of each model. *M.freq* is an estimate of the frequency of participants that is best explained by the model. (taking relative complexity into account). This is the estimate which was used to calculate the protected-exceedance probability. For the winning models, we also report pseudo $r^2$ (*p-r²*), a measure of the degree to which data are explained by the model, normalised for the data likelihood under chance[44]. This is calculated as $1-$(log-likelihood of the model for a participant/log-likelihood under chance), meaning that numbers below 0 indicate a worse fit than chance, and values close to 1 indicate a perfect fit. This measure is reported for the 25th, 50th and 75th percentiles of participants. For test-retest analysis parameters, we considered the parameter to be moderately reliable if they had an ICC(2,1) between 0.5 and 0.75 and had good reliability if it was above 0.75.

## Results
We follow the analysis plan stated in the preregistration (https://osf.io/rej74) unless otherwise stated. The results reported here replicate in large part the results of a pilot study with 54 participants (see Supplementary Note 2, pilot data).

### Model-agnostic results
We anticipated that participants would use the values associated with the choices and their outcomes to guide their decisions. As an initial test of this hypothesis, we fitted two hierarchical Bayesian logistic regression models to each task and block. The models included subject and stimulus as grouping variables with group-level intercepts and slopes. Below we report the results for the population-level effect; for full results, see Supplementary Tables S1–S8. The first model tested the effect of the values presented to the participants (value of the negative outcome and cost of action) and the effect of successes on the previous trial, finding that for both tasks, participants were more likely to make an active choice if the difference between value of the negative outcome (offer) and cost of action (cost; offer-cost) was high (Robber, Block 1: $b = 0.048$, 95% CI = [0.043, 0.053], $BF_{10} = 9.476\text{e} + 19$, Block 2: $b = 0.043$, 95% CI = [0.039, 0.048], $BF_{10} = 2.18\text{e} + 14$; Factory, Block 1: $b = 0.038$, 95% CI = [0.035, 0.042], $BF_{10} = 5.388\text{e} + 22$, Block 2: $b = 0.038$, 95% CI = [0.034, 0.041], $BF_{10} = 1.202\text{e} + 14$). As expected, these results indicate that participants took the values into account when making their choices. We did, however, not see conclusive evidence in favour or against the effect of the outcome of the previous trial on the probability to choose to act on the current trial (Robber, Block 1: $b = 0.031$, 95% CI = [-0.185, 0.242], $BF_{10} = 0.114$, Block 2: $b = 0.199$, 95% CI = [-0.015, 0.401], $BF_{10} = 0.597$; Factory, Block 1: $b = 0.195$, 95% CI = [0.031, 0.354], $BF_{10} = 1.314$, Block 2: $b = 0.221$, 95% CI = [0.033, 0.403], $BF_{10} = 1.377$). We did see an interaction effect between the difference between offers and costs, and the outcome of the previous trial in the robber task (Block 1: $b = 0.023$, 95% CI = [0.016, 0.031], $BF_{10} = 6.675\text{e} + 14$; Block 2: $b = 0.031$, 95% CI = [0.023, 0.038], $BF_{10} = 1.652\text{e} + 15$) whereby successes promoted active choices for high value offers (offer-cost), indicating that the outcomes affected choices. In the factory task, this effect was inconclusive in block 1 ($b = 0.006$, 95% CI = [0.002, 0.011], $BF_{10} = 0.101$), but evident in block 2 ($b = 0.015$, 95% CI = [0.01, 0.02], $BF_{10} = 7.773\text{e} + 15$).

However, this conceptualisation of learning does not consider the possibility that learning may only occur with repeated encounters with the same stimulus, and not between stimuli. This is a distinct possibility as participants were explicitly told that the probability of success changes with each stimulus. Therefore, the outcome of the last trial before a change in stimulus should not impact their choices with the next stimulus. As such, to

further test if participants learned from the outcomes of their choices, we constructed a regression model to test whether the probability of making an active choice changed over trials with the same stimulus and how that interacted with the underlying probability of success.

We found clear evidence in favour of an effect of the interaction between the number of trials with the same stimuli and the success probability associated with that stimulus in the robber task (Block 1: $b = 0.138$, 95% CI = [0.059, 0.218], $BF_{10} = 18.737$, Block 2: $b = 0.211$, 95% CI = [0.13, 0.292], $BF_{10} = 6.992e + 17$) whereby the more trials the participants encountered the stimuli with low success probability the fewer active choices were made. This was evident in the second block of the factory task ($b = 0.117$, 95% CI = [0.047, 0.186], $BF_{10} = 12.242$) but inconclusive in block 1 ($b = 0.078$, 95% CI = [0.007, 0.148], $BF_{10} = 0.337$). We also found a main effect of trial in the robber task (Block 1: $b = $ -0.106, 95% CI = [−0.152, −0.061], $BF_{10} = 6.012e + 15$; Block 2: $b = $ -0.16, 95% CI = [−0.208, −0.112], $BF_{10} = 4.457e + 15$) whereby participants had a higher probability to choose to act in early choices with a stimulus compared to the later ones, this effect was also inconclusive in the factory task in block 1 ($b = −0.065$, 95% CI = [−0.105, −0.025], $BF_{10} = 3.299$) but not in block 2 ($b = −0.082$, 95% CI = [−0.122, −0.043], $BF_{10} = 151.968$; for full results see Supplementary Tables S9–S16). These results show that participants learned from the outcomes of their active choice in the robber task, but the results for the factory task are inconclusive.

In a follow-up post-hoc analysis, which was not part of the preregistered protocol, we explored potential differences between the tasks. Participants chose the active option more frequently in the robber task than the factory task in both block 1 (Robber: $M = 0.661$, SD = 0.119; Factory: $M = 0.6140$, SD = 0.110; Wilcoxon signed-rank test: $V = 25961$, 95% CI = [0.033, 0.067], $p = 6.314e-11$) and block 2 (Robber: $M = 0.660$, SD = 0.136; Factory: $M = 0.604$, SD = 0.110; $V = 27815$, 95% CI = [0.050, 0.067], $p = 3.196e-16$). In contrast, the difference in the number of successes participants experienced when they chose the active option in the different tasks was not significant in the first (Robber: $M = 0.512$, SD = 0.077; Factory: $M = 0.502$, SD = 0.093, $V = 20886$, 95% CI = [−0.005, 0.023], $p = 0.221$) or second block (Robber: $M = 0.512$, SD = 0.0821; Factory: $M = 0.514$, SD = 0.080, $V = 19241$, 95% CI = [−0.014, 0.014], $p = 0.994$). We further did not see statistically significant differences in the mean number of successes between the two blocks of the robber task ($V = 19071$, 95% CI = [−0.013, 0.011], $p = 0.893$) or factory task ($V = 17356$, 95% CI = [−0.023, 0.005], $p = 0.22$).

### Choice behaviours are well captured by a Bayesian model

To further analyse participants' choice behaviour, and to test if participants' choice behaviour could be described by a model including a Bayesian prior, we constructed four different types of choice model. Initially, the models were fitted to each task and block separately.

Participants' choice behaviour in the robber task was robustly best fitted by the Bayesian learner model in both blocks (Block 1: PXP = 0.996;

Block 2: PXP = 1). This was only the case in the second block of the factory task (Block 1: PXP = 0.094; Block 2: PXP = 0.998). The best-fitting model in the first block of the factory task was the Static probability model (PXP = 0.906; see Table 2). These results are consistent with those from the regression models, indicating that learning was not as robust in the factory task as in the robber task (for how models differ in predictions of choices over trials, see Fig. 1E, F). Altogether, these results indicate that the Bayesian model provides the best overall account of participants' behaviour when modelling the tasks separately.

We next tested augmentations of the winning models, augmenting the Bayesian learner with between-stimulus updating and both the Static probability model and Bayesian learner with a bias parameter in stepwise model comparisons. Results showed that the choice data in the robber task was best fitted by the unaugmented Bayesian learner model (Block 1: $p\text{-}r^2$ 25th percentile = 0.428, median = 0.578, 75th percentile = 0.700; Block 2: 25th percentile = 0.437, median = 0.573, 75th percentile = 0.735) whereas the choice data in the factory task was best fitted by a model that included an action bias on top of the Static probability model in block 1 ($p\text{-}r^2$ 25th percentile = 0.443, median = 0.645, 75th percentile = 0.784) and the basic Bayesian model in block 2 ($p\text{-}r^2$ 25th percentile = 0.496, median = 0.691, 75th percentile = 0.817); for full results see Supplementary Note 1, Supplementary Results and Supplementary Table S17).

### The parameters of the prior correlate across tasks

As an initial test of task invariance, we tested whether the mean of the expected probability of success given action correlated between tasks when estimated using the best-fitting models from the stepwise model comparisons of the augmented models. For each task and block, we extracted each participant's parameters of the prior or the static subjective probability for block 1 of the factory task. In the first block, we found a strong correlation between the $\mu_0$ parameter of the prior in the robber task and the non-Bayesian counterpart, $p$, in the factory task ($r_s(277) = 0.363$, $R^2 = -0.455$, $p < 10e-4$; Fig. 2A). Similarly, we found strong correlations between the parameters of the priors in each task in the second block ($\mu_0$: $r_s(277) = 0.563$, $R^2 = -0.175$, $p < 10e-4$; $\sigma_0^2$: $r_s(277) = 0.395$, $R^2 = -0.045$, $p < 10e-4$; Fig. 2B, C).

### A task-invariant prior captures the participants' behaviour

To test the question of task invariance formally, we next constructed two different models that could capture the participants' behaviours in both tasks. Both models were constructed with separate SoftMax functions for each task, allowing for different levels of randomness in choice across tasks, and included an action bias parameter only in the factory task, as informed by the stepwise model comparisons performed separately for the two tasks (see Supplementary Table S17). Critically, the difference between the two models was that one assumed that participants shared the parameters of the prior across the tasks (1-prior model) while the other assumed separate priors in each task (2-prior model). As hypothesised and preregistered, participants' choice data were best fit by a model with a single prior for both tasks (Block 1: $M.Freq = 0.577$, $PXP = 0.995$; Block 2: $M.Freq = 0.690$, $PXP = 1$), demonstrating task invariance of the prior. As shown in Fig. 3, this model captures the observed behaviour well ($p\text{-}r^2$ for Block 1 at 25th percentile = 0.441, median = 0.583, and 75th percentile = 0.708; Block 2: 25th percentile = 0.458, median = 0.601, and 75th percentile = 0.741).

### The measured prior is reliable

We next investigated if the parameters of the prior were stable within the session and over time. The follow-up session consisted of only one block of each task with new stimuli, and choice data were modelled using the one prior model ($p\text{-}r^2$ at 25th percentile = 0.539, median = 0.644, and 75th percentile = 0.762). For model comparison between the one and two prior models in the follow-up session see Supplementary Table S18.

We quantified the stability of the prior by calculating interclass correlations (ICC) between the two blocks of the first session ($N = 279$) and between the blocks on the first day and the follow-up ($N = 101$). We found the stability of both parameters of the prior to range from moderate to good as indicated by

### Table 2 | Model comparison for basic models

| Block | Task | | SoftMax | Win-stay lose-shift | Bayesian learner | Static probability |
|---|---|---|---|---|---|---|
| Block 1 | Robber | M.freq | 0.1008 | 0.1368 | **0.4500** | 0.3125 |
| | | PXP | 0.0000 | 0.0000 | **0.9958** | 0.0042 |
| | Factory | M.freq | 0.1044 | 0.1458 | 0.3407 | **0.4092** |
| | | PXP | 0.0000 | 0.0000 | 0.0936 | **0.9064** |
| Block 2 | Robber | M.freq | 0.0968 | 0.1360 | **0.5608** | 0.2065 |
| | | PXP | 0.0000 | 0.0000 | **1.0000** | 0.0000 |
| | Factory | M.freq | 0.0731 | 0.1339 | **0.4754** | 0.3176 |
| | | PXP | 0.0000 | 0.0000 | **0.9985** | 0.0015 |

Bold text signals the best-fitting model in this model comparison.

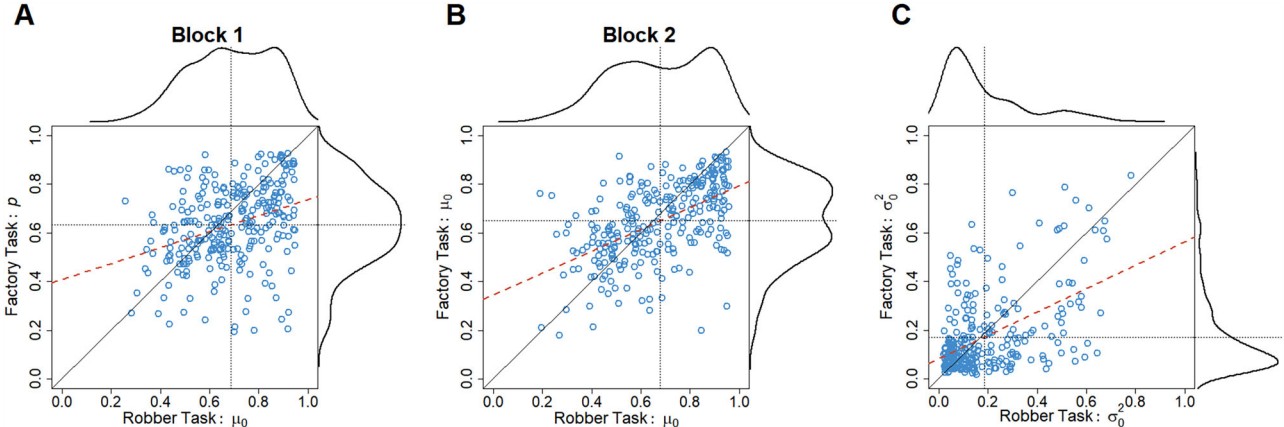

**Fig. 2 | Correlations between parameters of the prior across tasks.** Scatter plots show each participant's extracted parameters for each task and block (N = 279). The dotted lines indicate the means of the parameters, and the diagonal indicates a perfect correlation. The red dashed lines are regression lines. Distributions above and to the right of the scatter plots show the rule-of-thumb bandwidth Gaussian kernel density estimates of the corresponding parameter values implemented by default in R (0.9 times the minimum of the standard deviation and the interquartile range divided by 1.34 times the sample size to the negative one-fifth power). Reported p values from the permutation test with 10,000 permutations. **A** Participants' parameter values for the $\mu_0$ and p parameters in the first block of the robber and factory task, respectively. **B** Participants' parameter values for the $\mu_0$ parameter in the second block of the robber and factory task. **C** Same as b but for the $\sigma_0^2$ parameter.

ICC (within session: $\mu_0$ ICC(2,1) = 0.691, 95% CI = [0.625, 0.748], $\sigma_0^2$ ICC(2,1) = 0.591, 95% CI = [0.507, 0.663]; between sessions, Block 1 to follow-up: $\mu_0$ ICC(2,1) = 0.748, 95% CI = [0.648, 0.822], $\sigma_0^2$ ICC(2,1) = 0.645, 95% CI = [0.516, 0.746]; Block 2 to follow-up: $\mu_0$ ICC(2,1) = 0.746, 95% CI = [0.645, 0.821], $\sigma_0^2$ ICC(2,1) = 0.652, 95% CI = [0.525, 0.751], see Fig. 4C, F). These data demonstrate that the prior that we quantify with our method is reliably measured and stable. Alternative reliability measures are reported in Supplementary Tables S19–S21 and support the same conclusion.

## Questionnaire analysis

We next investigated if the parameters of the prior were correlated with positive affect and nonspecific negative affect. Because the overall best-fitting model was the 1-prior model, the parameters of each participant's prior were estimated with this model. Participants' estimated $\mu_0$ and $\sigma_0^2$ were then correlated with their total PANAS-P and STAI-T scores using Spearman's correlation ($r_s$). Figure 5A shows the correlations for the parameters of the prior in blocks 1 and 2 and their relation to the total scores of the questionnaires.

As hypothesised and preregistered, we found that $\mu_0$ of the prior was positively correlated with positive affect in block 1 ($r_s(277) = 0.120$, $p = 0.046$; Fig. 5B, top), indicating that participants with a more optimistic prior also had higher levels of self-reported positive affect. This was replicated in the follow-up session ($r_s(99) = 0.225$, $p = 0.023$; Fig. 5B, bottom), but the correlation in block 2 on the first day was not statistically significant ($r_s(277) = 0.028$, $p = 0.643$). However, when, in accordance with our pre-registration, we removed three participants who failed more than two of the six additional attention-checks present in the questionnaires in the first session, the correlation in block 1 was no longer statistically significant ($r_s(274) = 0.118$, $p = 0.051$, see the red marks in Fig. 5B, top for highlight of removed participants). We note, however, that the strength of the association was not affected, and the change in significance is a consequence of decreased power. Surprisingly, we did not see the hypothesised correlation between the $\sigma_0^2$ of the prior and STAI-T (Block 1: $r_s(274) = 0.016$, $p = 0.792$; Block 2: $r_s(274) = 0.102$, $p = 0.092$; Follow-up: $r_s(99) = 0.003$, $p = 0.978$).

## Discussion

We developed two different aversively-framed decision-making tasks in which executing costly actions might lead to the avoidance of potentially negative outcomes. In line with our preregistered hypotheses, the tasks were amenable to being modelled using an augmented Bayesian observer model that includes a prior belief about the probability of success of executing such actions. As hypothesised and preregistered, our results demonstrate that this prior belief is rather task-invariant and reliable when measured on two different occasions and is positively correlated with positive affect, providing convergent validity of the measured prior, and supporting the interpretation that a generalising pessimistic prior about active avoidance may play an important role in depression.

Our data demonstrate that we can quantify a task-invariant prior belief about the probability of success when performing an action to avoid potentially negative outcomes. Task invariance is a key property of the measured prior and a prerequisite for it to be relevant for behaviour observed outside of the laboratory. To achieve task invariance, we followed the strategy previously used by Houlsby et al. [16] and designed two decision-making tasks which admit a potential prior belief about the controllability of the environment, that is, the probability of success when performing an action to avoid a potential negative outcome. The two tasks differ in two important ways: the framing of the decisions and the valence of the outcomes. These differences ensured that the measured prior does not merely reflect task-specific behavioural strategies.

The fact that the prior was task-invariant supports a key assumption of the Bayesian account of helplessness, namely generalisation of expectations about controllability [2,3]. Previous theoretical work has suggested that learned helplessness results from learning a pessimistic prior about the success of one's actions when performing them in an uncontrollable environment. This prior then instils the expectation of failure when performing an action, and as the prior generalises the lack of controllability across different contexts, helplessness ensues [2,3]. Although this theoretical account is very suggestive, empirical evidence for the proposal has been scarce. Recent studies have found evidence that prior beliefs, as conceptualised in Bayesian theories, relate to traits such as optimism [45,46], behavioural activation [13] or expressions of symptoms of depression and anxiety [15]. However, these studies only employed single behavioural tasks, and so could not test the existence of priors that generalise broadly [17,18]. By demonstrating task invariance, our data fills a missing knowledge gap and provide support for the Bayesian account of helplessness.

We operationalised the estimate of controllability as a prior belief about the probability of success when performing an action to avoid negative outcomes. Ligneul and colleagues [47] have recently suggested an alternative, which assesses controllability by comparing the predictive accuracy of two internal models of the environment: an "actor" that includes the effects of the agent's actions, and a "spectator" that considers the environment as evolving independently of those actions. By thresholding this dynamic

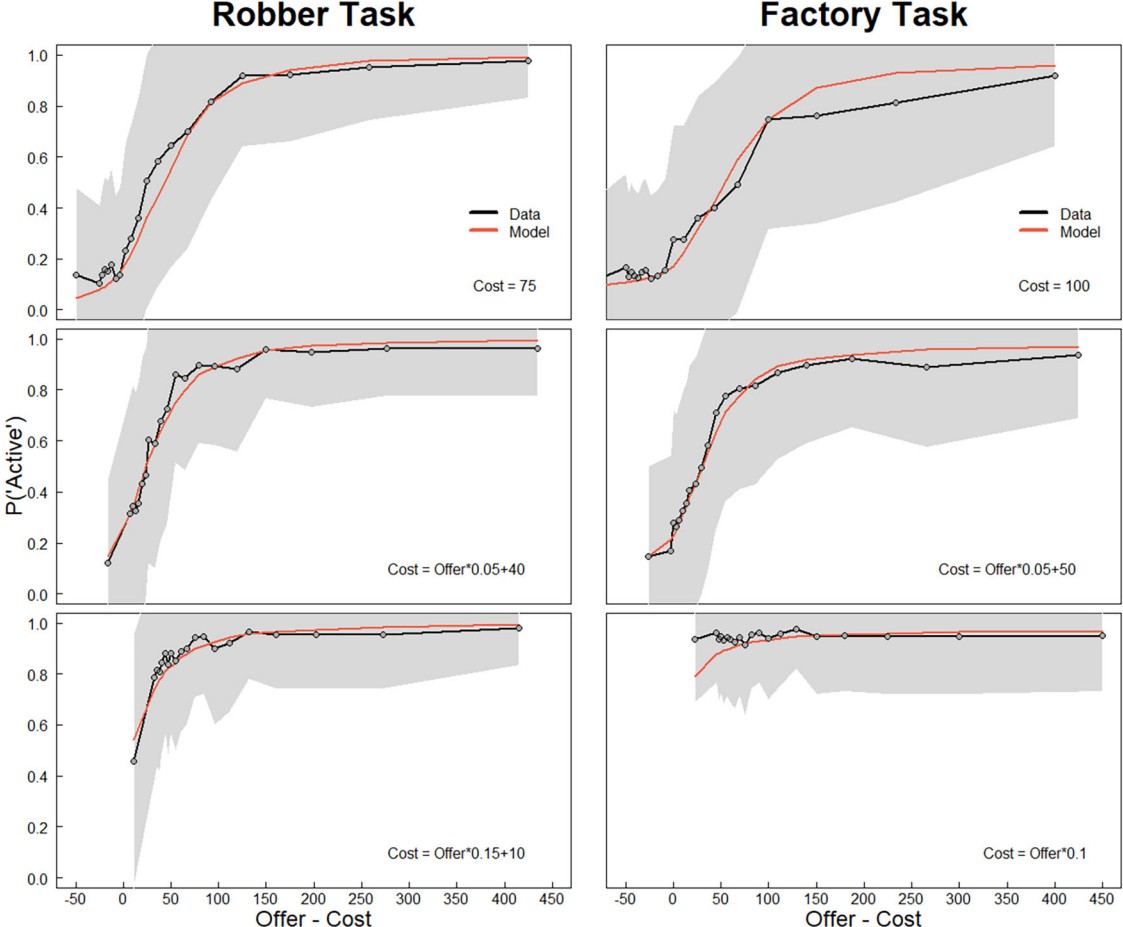

**Fig. 3 | Model prediction of task-invariant prior.** Red line shows model behaviour simulated with the participants' parameter values and design matrix (the specific order of offers, costs, success probabilities and number of trials with a stimulus) with the winning 1-prior model with a shared prior across tasks in the first block (mean across 1000 simulations). Black lines (and grey shading) show observed mean (and standard deviation) of behaviour for each task in the first block and cost function.

estimate, the model can arbitrate between the two models when making choices in a prediction task that depend on the level of inferred controllability. Akin to learned helplessness, they showed that humans, after experiencing uncontrollable shocks in a separate environment, increase the threshold parameter in the model. In turn, this results in greater reliance on the spectator model in subsequent settings. One important feature of Ligneul's model and experimental paradigm is that it disambiguates controllability and uncertainty, two quantities normally conflated in previous paradigms studying the impact of controllability in decision-making[20]. Although this is an important feature of the experimental paradigm which we did not control, this does not impact our estimate of the prior that ultimately reflects an expectation of controllability of the environment. This is because we do not compare two different conditions differing in controllability, but the estimated level of expected controllability of the environment in general. In fact, we would expect a positive correlation between the threshold in Ligneul's model and the prior beliefs measured here.

When modelling the tasks separately, we found very strong evidence for a full Bayesian model using a prior belief about the probability of success when performing an action to avoid potentially negative outcomes for both blocks of the robber task and the second block of the factory task. Although this full Bayesian model was outperformed by the simpler Static probability model in the first block of the factory task, this model can be seen as a heuristic implementation of the full Bayesian model without updating. One possible reason for this apparent lack of updating in the factory task is that participants chose the active action to a lesser extent in this block of the task compared to the robber task and the second block. This thereby provided fewer opportunities to sample participants' potential learning in this block.

This difference in the tendency to choose the active option may stem from the fact that the factory task lies in the negative domain in which loss aversion predominates[48] in a way that might potentially initially increase the difficulty of processing and calculating the value difference.

A crucial aspect of our results is that we found this prior to be reliable, both across blocks on the same test occasion and on separate testing occasions. Here we find that the degree of reliability of the prior mirrors that of the PANAS questionnaire. Previous research investigating the reliability of parameters from computational models have generally found low test-retest reliability of these parameters[49–51]. The reliability of cognitive measures can be increased by constructing latent variables using factor analysis that capture the shared variance across multiple measures of the same construct[52]. In this sense, the use of two different tasks that could depend on the same prior may boost the reliability of the measured prior. While low reliability has been attributed to dynamic characteristics changing over time[50,51], the reliability of the parameters of our priors indicates that these might be related to more stable traits. However, dynamic characteristics of the constructs are not sufficient to explain low reliability in all cases, as it can be boosted by experimental design[53,54]. Furthermore, while the prior does not shift substantially between the sessions or tasks, we would not expect that the observed prior is static; rather, it should evolve over time to reflect one's life experiences. However, if the prior reflects some key experiences with roots in childhood, we should not expect it to change much over the course of days. One possibility that we have not tested here is that the prior is informed by what amounts to a higher-order prior that shifts more slowly or is not affected by the tasks employed, much like the hierarchical structure employed in volatility learning[55].

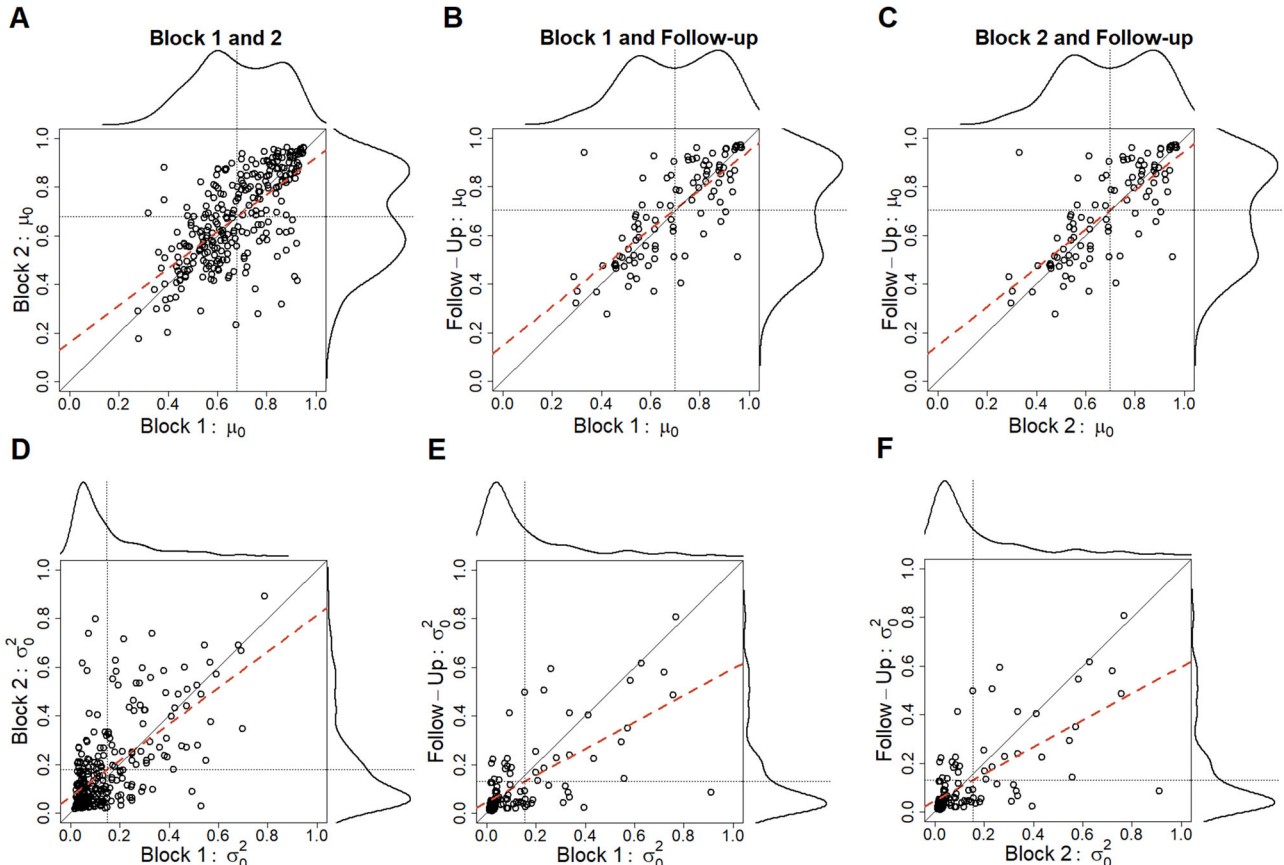

**Fig. 4 | Test-retest reliability for the prior fitted across tasks.** Each point in each plot is the estimated parameter value for a participant at two different time points, the dotted lines show the mean of the parameter at the different time points, and the diagonal indicates a perfect correlation. The red dashed lines are regression lines. Distributions above and to the right of the scatter plots show the rule-of-thumb bandwidth Gaussian kernel density estimates of the corresponding parameter values implemented by default in R (0.9 times the minimum of the standard deviation and the interquartile range divided by 1.34 times the sample size to the negative one-fifth power). **A** Correlation between the task-invariant $\mu_0$ parameter in blocks 1 and 2 ($N = 279$). **B** Correlation between the task-invariant $\mu_0$ parameter in block 1 and the same parameter 1 week later ($N = 101$). **C** Correlation between the task-invariant $\mu_0$ parameter in block 2 and the same parameter 1 week later ($N = 101$). **D–F** Same as **A–C** but for the $\sigma_0^2$ parameter.

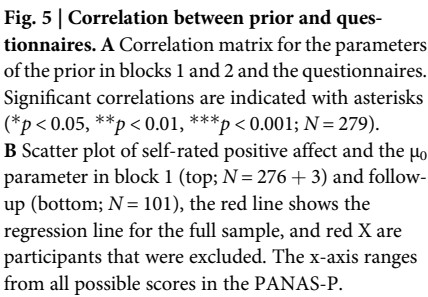

**Fig. 5 | Correlation between prior and questionnaires. A** Correlation matrix for the parameters of the prior in blocks 1 and 2 and the questionnaires. Significant correlations are indicated with asterisks (*$p < 0.05$, **$p < 0.01$, ***$p < 0.001$; $N = 279$). **B** Scatter plot of self-rated positive affect and the $\mu_0$ parameter in block 1 (top; $N = 276 + 3$) and follow-up (bottom; $N = 101$), the red line shows the regression line for the full sample, and red X are participants that were excluded. The x-axis ranges from all possible scores in the PANAS-P.

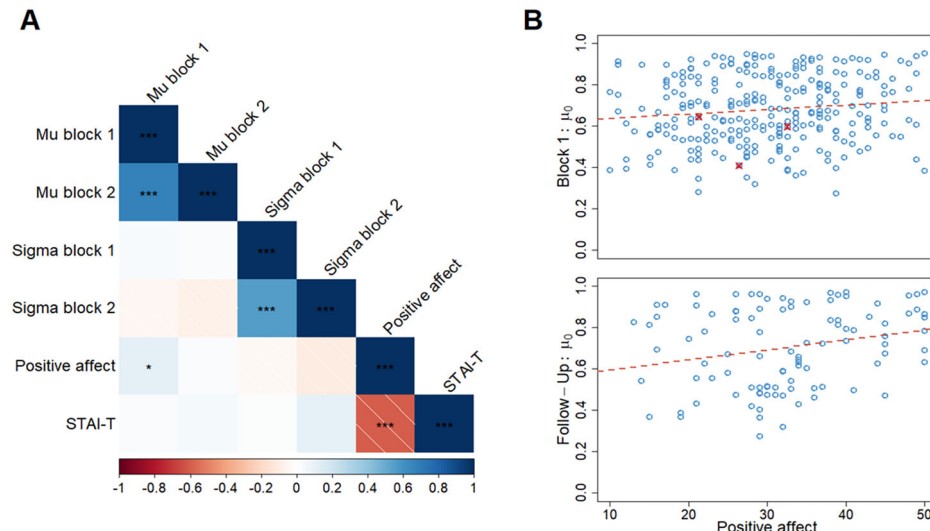

## Limitations

A limitation is that we did not augment the Static probability model with learning, as in a stimulus-specific or continuous Q-learning model, because simulations showed that this model was not identifiable against the basic Bayesian learner and the static probability model in our tasks. While there are key differences between the Bayesian learner and a Q-learning model, notably fixed versus changing learning rates, in some cases a Bayesian learner can come to implement a near-fixed learning rate if the underlying

generative characterisation of the environment allows the latent variable that needs to be estimated (in our case: success probability) to change over time[56]. As such, more specific manipulations may be needed to distinguish these models and until such time, "priors shared across tasks" is equivalent to "initial estimates and learning rates are shared across tasks".

While the correlation between positive affect and the mean of the prior adds some credence to the link between the prior and depressive symptoms, the correlation was rather weak, and the association fails to cross the threshold for statistical significance when participants who failed attention checks were removed. Positive affect was hypothesised to be associated with the prior, since reduced positive affect is central in depression[24,25,31,32] and a promising target for treatment[31,57,58]. Furthermore, positive affect has a large variance in the general population[26], which we therefore used as a proxy for depressive symptoms in a non-clinical sample. This weak association may indicate that the mean of the prior is not strongly related to depression symptoms. However, it may also mean that positive affect is an imprecise measure of anhedonia, which has itself been suggested to be closely related to expectations of controllability of the environment[59]. We hypothesise that anhedonia would be more closely linked to the measured prior. However, the variance of anhedonia in a non-clinical sample as the one of this study is expected to be low compared with clinical populations[60]. Future studies should examine the relationship between our prior and anhedonia in a clinical sample of depressed patients.

## Conclusions

Our data show that individual differences in the tendency to act to avoid a negative outcome can be conceptualised as a generalised prior belief in a Bayesian framework, providing support to theoretical accounts of learned helplessness and depression. Future research should explore how this prior is related to apathy, experimental manipulation of controllability and patient groups and treatment outcomes.

## Data availability

All anonymized data used in the analysis is openly available at zenodo.org https://doi.org/10.5281/zenodo.13745010.

## Code availability

All code (analysis, simulation and experiment code) are openly available at the lead author's GitHub (https://github.com/Granwald/healthyPriors) and OSF (analysis and simulations: https://doi.org/10.17605/OSF.IO/P6GDY; experiment code: https://doi.org/10.17605/OSF.IO/8GBSU).

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

## Acknowledgements

This work was supported by a grant awarded by the Marianne and Marcus Wallenberg Foundation to M.G.M. (MMW 2020-0013). P.D. is funded by the Max Planck Society and the Humboldt Foundation. P.D. is a member of the Machine Learning Cluster of Excellence, EXC number 2064/1 – Project number 39072764 and of the Else Kröner Medical Scientist Kolleg "ClinbrAIn: Artificial Intelligence for Clinical Brain Research". The funders have/had no role in study design, data collection and analysis, decision to publish or preparation of the manuscript.

## Author contributions

Conceptualisation: T.G. and M.G.M.; Funding acquisition and supervision: M.G.M.; Investigation and writing—original draft: T.G.; Writing—review & editing and methodology: T.G., P.D., M.L. & M.G.M.

## Funding

## Competing interests

The authors declare no competing interests.
