## [Transparent Peer Review file · Communications Psychology]

A task-invariant prior explains trial-by-trial active avoidance behaviour across gain and loss tasks

Corresponding Author: Mr Tobias Granwald

Version 0:

Decision Letter:

Dear Mr Granwald,

Thank you for your patience during the peer-review process. Your manuscript titled "Bayesian Priors in Active Avoidance" has now been seen by 2 reviewers, and I include their comments at the end of this message. They find your work of interest but raised some important points. We are interested in the possibility of publishing your study in Communications Psychology, but would like to consider your responses to these concerns and assess a revised manuscript before we make a final decision on publication.

We therefore invite you to revise and resubmit your manuscript, along with a point-by-point response to the reviewers. Please highlight all changes in the manuscript text file.

Editorially, we consider it important that the revised manuscript addresses the methodological questions that the reviewer raised, such as the use of a measure that takes model complexity into account for model comparison. Please ensure that the revised version includes a link to the preregistration that allows the reviewers to access the preregistration without registering their identity. In line with our detailed preregistration policies (<https://www.nature.com/commpsychol/editorial-policies/preregistration-policy>), we ask that you disclose all deviations from the preregistered protocol (if any) and explain the rationale for deviation (e.g., flaw, feasibility, suboptimality). In cases of deviation from the preregistered analysis plan for reasons other than fundamental flaw or feasibility, the originally planned analyses must also be reported. Please also improve the clarity of the display items, as suggested by Reviewer #1. More information about journal style and guidance for the presentation of the Results is included in the checklist.

I am attaching an Editorial Requests Table that details critical reporting requirements for the revised manuscript. Please attend to each item and ensure your manuscript is fully compliant. If your revised manuscript is not aligned with these requests on major issues, such as those concerning statistics, it may be returned to you for further revisions without re-review.

Please submit the following items:

- Revised manuscript
- Point-by-point response to the referees' comments
- Cover letter (as a separate document)
- <https://www.nature.com/documents/nr-reporting-summary.zip>>Nature Research Reporting Summary
- <https://www.nature.com/documents/nr-editorial-policy-checklist.pdf>>Editorial Policy Checklist

- Completed Editorial Request Table (attached).

via this link: Link Redacted .

Additional guidance is available in our style and formatting guide Communications Psychology formatting guide.

Best regards,

Troby Lui

Troby Lui, PhD
Associate Editor
Communications Psychology

REVIEWER EXPERTISE:

Reviewer #1: computational psychiatry, modelling

Reviewer #2: computational psychiatry, modelling

REVIEWER REPORTS:

Reviewer #1 (Remarks to the Author):

Thank you for the opportunity to review the manuscript titled "Bayesian Priors in Active Avoidance" for Communications Psychology. The manuscript investigates avoidance across two behavioral tasks, using data from a large sample of online participants, combined with Bayesian modelling. Crucially, the two behavioral tasks differ in so far as one task operationalized active avoidance of losing a win, whereas the other task investigates avoidance of a loss. In computational terms, the participants' tendency for active avoidance was best captured by a task-invariant prior, which correlated with positive affect and was relatively stable in a follow-up experiment.

I find this work very interesting, and I would like to start by pointing out several methodological and conceptual strengths:

P1. It uses two behavioral tasks that are designed in a way to test different sides of active avoidance, which reflects a departure from single-task studies with oftentimes limited generalizability.

P2. The application of computational models is solid and reflects good scientific practice, including a parameter recovery exercise and model recovery approaches.

P3. The language is clear and concise.

P4. Hypotheses and analyses follow the pre-registered scheme (although I could not access the pre-registration upon the time of reviewing the manuscript – would it be possible to make it available?)

P5. The conclusion of a stable, task-invariant prior on active avoidance is furthermore supported by a follow-up study that investigated the reliability of the prior over time in a random subset of participants.

P6. The task-invariant prior also showed some interesting correlations with psychological self-report variables such as affectivity.

Below please find some suggestions on how to improve on the clarity of the manuscript and the computational modelling:
Introduction

S1. Regarding the computational foundations of optimism, the work by E. Fisher and J. Hohwy might be interesting and relevant in the context of the manuscript: Fisher, Whyte & Hohwy, 2024, OSF preprints; Fisher & Hohwy, 2024, Entropy. This could help with some inspiration on the connection between Bayesian theories and optimism, which I find a bit generic here (L. 46: the initial prior, and previous experiences [...])

S2. It feels like the psychological constructs in the introduction are related, but only loosely (behavioral activation,

avoidance, optimism, learned helplessness) – I think the intro might benefit from focusing on one or two more tightly interconnected concepts, i.e., avoidance and learned helplessness

S3. L. 110 I don't think the numbering of the hypotheses is necessary here unless they would also be referred to later in the text

Results section

Model agnostic results

S4. L. 146 grammatical issue? Rephrasing might be helpful here

S5. For the Bayesian logistic regression, it would be interesting to report some measures of model convergence (trace plots, effective sample size (ESS), R-hat) in the supplements.

S6. How were the priors defined (I am assuming uniform/ Gaussian?) and how sensitive are the modelling results to different choices of priors (i.e., different choices of mean and SD)?

S7. Another thing to check would be the distribution of the residuals to ensure the results are trustworthy

S8. Generally: I suggest adding some descriptive results about task performance as part of the model-agnostic results (mean/ range of wins or losses across participants, ...)- this may help to understand the special situation in Block 1 of the factory task?

Figure 1

S9. C+D: it took me quite a while to understand these plots. The Y-axis label is not descriptive enough – suggestion: value of choice or sth similar. Also, pls describe the meaning of the dots in the legend (I am assuming that these were the possible sizes of the offers) –

S9.1 Question: why did you decide to have many low offers in the robber task and more high offers in the factory task instead of distributing it uniformly? Was this a decision made based on pilot data? Would be good to explain this choice.

S10. E+F: These plots would benefit from a legend explaining the meaning of the different colors. I understand that these are shown in G but I think this is a bit too indirect. I would recommend a heatmap-type legend here, with the colors ranging from blue to red and representing the different success probabilities from 7 to 92%

S11. I do not quite understand the meaning of the opacity of the line. Could you rephrase the sentence in the legend (L. 175, is quite a convoluted sentence), and add a remark about this in the figure itself as well? My goal with this comment is to improve the first-glance interpretability of the plots.

Bayesian models

S12. Four different choice models were developed. They are described in more detail in the supplements, but it would be helpful to have a tabular (or graphical) overview of the models also in the main text, especially displaying what their choice function looks like and how they differ from one another in mathematical terms.

S13. I am assuming that you used uninformative priors for all models, correct? How sensitive are the modelling results when choosing different parameters for the priors?

S13.1. (Reproducibility of the results is a bit hard to judge bc access to code is not granted yet)

S14. Can you make the motivation/ conceptual thinking for the model augmentations, i.e., the action bias a bit clearer? L. 275

S15. L. 342, legend of Fig. 2: what does design matrix refer to here? Please specify

S16. Fig. 4: I think Block 1 and 2 in the follow up experiment refer to the different tasks, correct? Or do the plots display prior parameters across tasks? Please specify in title or so from what task(s) the data come.

S17. The model comparison is performed at the group-level using protected-exceedance probabilities. This allows some conclusions about which model is the most likely at the population level. This approach does not balance between goodness of fit and model complexity, like classical information criteria. I recommend evaluating the models with a measure that takes model complexity into account.

S18. Have you considered reporting an absolute measure of goodness of fit, at least for winning models – akin to e.g. R²?

Discussion

L. 414-425, I find the argumentation here very convincing and agree that the authors, with their design have demonstrated an interesting way of studying Bayesian computational markers in behavioral experiments!

Limitations

S19. A clear limitation to the interpretations of a robust prior for avoidance that requires addressing is the less robust behavioral effect in the factory task, with many hypothesized effects showing up only in the second block – the reasons for this would need to be discussed (slower learning but why?) and I would appreciate some ideas on what happened in Block 1, whether these tasks maybe differed in other, unrelated aspects: difficulty? Motivation? The best fitting model seemed to be a static model with an overall action bias for Block 1 in the factory task.

S20. The correlation between positive affect and prior μ is relatively tender (Fig. 5a&b) – interpret with care esp. with respect to the implications for affective disorders

Methods

S21. Could you report some additional sample characteristics? I.e., mean completion time, number and range of prolific submissions, age, nationality

S22. How many attention checks were there in total?

Minor language issues

- L. 208: played with the low probability stimuli – suggest to say something like: encountered the stimuli with low success rates, to keep it consistent throughout.

- L. 236 too much italics

- L. 414: Replace “That” with “The fact that”

Best wishes,

ALE

Reviewer #2 (Remarks to the Author):

Thank you for the opportunity to review this manuscript. The authors have leveraged an innovative experimental design to determine the extent to which prior beliefs about the outcomes of actions are generalisable across different contexts. Although previous researchers have examined the influence of prior beliefs on decision making, these have been restricted to single task designs that are unable to determine whether the influence of individual differences in prior beliefs generalises across different tasks. By using two distinctive, aversively framed tasks to examine the influence of prior beliefs about the probability of success when performing an action to avoid potentially negative outcomes, the authors convincingly show that participants' prior beliefs reliably influenced decision making across different contexts. Importantly, by providing evidence of the convergent validity of the measured prior by demonstrating its correlation with self-report measures of positive affect, the authors results point to a Bayesian conceptualisation of helplessness, which has clear implications for our understanding of depression - namely, their results suggest that depressed states may be characterised by a generalisable pessimistic prior favouring active avoidance across different contexts.

In my view, the authors' submission is convincing, innovative and elegant, and fills an important gap in the literature. I am happy to recommend it for publication, in its current form, and I wish them all the best for their future endeavours.

Communications Psychology is committed to improving transparency in authorship. As part of our efforts in this direction, we are now requesting that all authors identified as 'corresponding author' create and link their Open Researcher and Contributor Identifier (ORCID) with their account on the Manuscript Tracking System prior to acceptance. ORCID helps the scientific community achieve unambiguous attribution of all scholarly contributions. You can create and link your ORCID from the home page of the Manuscript Tracking System by clicking on 'Modify my Springer Nature account' and following the instructions in the link below. Please also inform all co-authors that they can add their ORCID to their accounts and that they must do so prior to acceptance.

Version 1:

Decision Letter:

Dear Mr Granwald,

Your manuscript titled "Bayesian Priors in Active Avoidance" has now been seen by our reviewers, whose comments appear below. In light of their advice I am delighted to say that we are happy, in principle, to publish a suitably revised version in Communications Psychology.

We therefore invite you to revise your paper one last time to address the remaining concerns of our reviewers and a list of editorial requests. At the same time we ask that you edit your manuscript to comply with our format requirements and to maximise the accessibility and therefore the impact of your work.

EDITORIAL REQUESTS:

SUBMISSION INFORMATION:

In order to accept your paper, we require the files listed here <https://www.nature.com/documents/commsj-file-checklist.pdf> .

OPEN ACCESS:

* DATA AVAILABILITY:

Link Redacted

Best regards,

Troy Lui

Troy Lui, PhD
Associate Editor
Communications Psychology

REVIEWERS' COMMENTS:

Reviewer #1 (Remarks to the Author):

Thank you for the opportunity to review a revised version of the manuscript on Bayesian Priors and Avoidance for Comms Psych. The authors have meticulously revised their work based on my previous comments. Their dedication makes me feel fortunate to be a part of such a diligent and committed crowd. The updates include a more concise introduction, more thorough descriptions of the computational models as well as additional measures of fit, updated legends and revised figures with improved readability.

After re-analyzing the dataset without 3 participants who failed the attention checks, the correlation between the prior μ and positive affect was no longer significant. I am unsure how convincing I find the argument that this is due to a loss of power. However, since the authors now also discuss the implications of this correlation much more carefully in the discussion section, I think that this is acceptable. I suggest updating the sentence in the abstract about this correlation (manuscript p. 1 l. 28-29), since many readers will probably only read the abstract and should get more context about this finding. Other than that, I am happy to recommend this interesting and well-executed manuscript for publication.

Reviewer #2 (Remarks to the Author):

Although my own review required no further modifications to the manuscript, it is clear from the authors' extensive responses to the first reviewer that they have taken this feedback seriously and gone to considerable lengths to address any concerns about the suitability of this manuscript for publication. Once again, I applaud the authors' efforts, recognise the unique and meaningful contribution of their research to the literature, and am happy to recommend its publication in Communications Psychology.

Thank you for giving us the opportunity to resubmit a revised version of our manuscript entitled ‘Bayesian Priors in Active Avoidance’. We found the suggestions and comments of the reviewers most helpful and constructive and have addressed them thoroughly as detailed below.

The comments from the reviewers are provided in “cited” plain text and our responses are provided in **bold**. Additions made in the manuscript are shown with **red text** when whole sentences or paragraphs have been added and **red and underlined** is used to highlight parts of sentences that have been added. In order to comply with the formatting guidelines of *Communications Psychology* we have reordered the manuscript so the method now comes before the results. As a consequence, method descriptions in the results section that were judged superfluous were removed. We therefore took the liberty of reordering the comments to follow the reordered manuscript.

We have also made other changes that were not directly prompted by the reviewers. These were changes in the formatting of the equations to more closely follow established conventions and a change in the results of the correlation between the mu of the prior and positive affect. The reason for this change was that when going through the scripts we found that 3 participants that failed more than 2 attention-checks that were present in the questionnaires had not been removed for this analysis. Following this change the results have been updated and the correlation between the mu of the prior and the positive affect was no longer significant at our selected alpha level after removal of these subjects. Please note that, since none of these subjects were outliers in the positive affect scale, the change in significance is attributed to a loss of power. Both the previous results and these new ones are provided in the Results section, and the discussion of the results has been updated.

The following section was included in the Results section:

However, when, in accordance with our preregistration, we removed 3 participants who failed more than 2 of the 6 additional attention checks present in the questionnaires in the first session, the correlation in block 1 was no longer statistically significant ($r_s(274) = 0.118, p = 0.051$, see the red marks in Fig. 5b, top for highlight of removed participants). We note, however, that the strength of the association was not affected, and the change of significance is a consequence of decreased power.

Reviewer comments and responses:

Reviewer #1:

“Thank you for the opportunity to review the manuscript titled “Bayesian Priors in Active Avoidance” for Communications Psychology. The manuscript investigates avoidance across two behavioral tasks, using data from a large sample of online participants, combined with Bayesian modelling. Crucially, the two behavioral tasks differ in so far as one task operationalized active avoidance of losing a win, whereas the other task investigates avoidance of a loss. In computational terms, the participants’ tendency for active avoidance was best captured by a task-invariant prior, which correlated with positive affect and was relatively stable in a follow-up experiment.

I find this work very interesting, and I would like to start by pointing out several methodological and conceptual strengths:

P1. It uses two behavioral tasks that are designed in a way to test different sides of active avoidance, which reflects a departure from single-task studies with oftentimes limited generalizability.

P2. The application of computational models is solid and reflects good scientific practice, including a parameter recovery exercise and model recovery approaches.

P3. The language is clear and concise.

P4. Hypotheses and analyses follow the pre-registered scheme (although I could not access the pre-registration upon the time of reviewing the manuscript – would it be possible to make it available?)

P5. The conclusion of a stable, task-invariant prior on active avoidance is furthermore supported by a follow-up study that investigated the reliability of the prior over time in a random subset of participants.

P6. The task-invariant prior also showed some interesting correlations with psychological self-report variables such as affectivity.

Below please find some suggestions on how to improve on the clarity of the manuscript and the computational modelling:”

We thank the reviewer for these positive comments, their thorough reading of our manuscript, and the insightful suggestions presented below. The preregistration is now openly available at: <https://osf.io/rej74>

Introduction

“S1. Regarding the computational foundations of optimism, the work by E. Fisher and J. Hohwy might be interesting and relevant in the context of the manuscript: Fisher, Whyte & Hohwy, 2024, OSF preprints; Fisher & Hohwy, 2024, Entropy. This could help with some inspiration on the connection between Bayesian theories and optimism, which I find a bit generic here (L. 46: the initial prior, and previous experiences [...])

We thank the reviewer for this comment and recommendation of references. After reading through the references we have carefully considered adding them to the Introduction. However, after revising the Introduction to make it more focused on learned helplessness and avoidance, as suggested below, we believe the Fisher & Hohwy, 2024, Entropy

reference is better suited in the discussion where other studies, which find evidence for prior beliefs, are presented. See the sentence below: Recent studies have found evidence that prior beliefs, as conceptualised in Bayesian theories, relate to traits such as optimism (Stankevicius et al., 2014, Fisher & Hohwy, 2024)...

“S2. It feels like the psychological constructs in the introduction are related, but only loosely (behavioral activation, avoidance, optimism, learned helplessness) – I think the intro might benefit from focusing on one or two more tightly interconnected concepts, i.e., avoidance and learned helplessness”

We thank the reviewer for this suggestion. We have now removed the more loosely connected concepts to try to tighten up the introduction. We here followed your suggestion to increase the focus on avoidance and learned helplessness.

“S3. L. 110 I don’t think the numbering of the hypotheses is necessary here unless they would also be referred to later in the text”

We thank the reviewer for this suggestion. As we only reuse these numbers in two places at the end of the methods section, we can see that it does not add much to keep the numbering of the hypotheses. Furthermore, in keeping with the formatting guidelines (Introduction “Should not contain summary of the results”) this section has been removed, and the numbering of the hypotheses has been removed from the methods section.

Methods

“S21. Could you report some additional sample characteristics? I.e., mean completion time, number and range of prolific submissions, age, nationality”

We thank the reviewer for this suggestion. We have now added a section in the supplementary material with all available sample characteristics beyond the material that had been available in the methods section. Now the following is added to the Methods section: (see Supplementary methods for full breakdown of sample characteristics).

And the following paragraph has been added to the Supplementary material: Below is a full breakdown of the sample characteristics of the 279 participants who participated in the current study. The median completion time of the participants in the first session was 1 hours and 34 minutes, the median completion time for the follow-up session was 1 hour. To participate, participants had to be above the age of 18 and below the age of 40. Complete self-reported demographic data were collected for all but 4 participants. The final sample for the first day had a mean age of 27.12 ($SD = 5.06$) years with 133 females and 142 males and 36 different nationalities reported: Australia (N = 1), Bulgaria (N = 1), Canada (N = 3), Chile (N = 8), China (N = 2), Croatia (N = 1), Czech Republic (N = 2), Estonia (N = 1), Finland (N = 2), France (N = 1), Germany (N = 1), Greece (N = 8), Hungary (N = 10), Israel (N = 3), Italy (N = 11), Latvia (N = 3), Mexico (N = 17), Netherlands (N = 4), New Zealand (N = 1), Nigeria (N = 2), Norway (N = 1), Philippines (N = 1), Poland (N = 38), Portugal (N = 40), Russian Federation (N = 1), Serbia (N = 1), Slovenia (N = 2), South Africa (N = 88), Spain (N = 2), Sweden (N = 3), Turkey (N = 2), Ukraine (N = 1), United Kingdom (N = 11), United States (N = 1), Vietnam (N = 1), Zimbabwe (N = 2). Participants’ mean number of previous approved submissions on prolific was 234.88 ($SD = 248.86$, median = 176, min = 0, max = 2559)

“S22. How many attention checks were there in total?”

We apologize for this oversight in our reporting and thank the reviewer for pointing it out. In total there were eight attention checks, made up of two in each block. One of these was randomly placed within the task between two stimuli, and one directly after the end of the instructions provided before the start of the block. This is now clearly stated in the Methods section. See the underlined part of the text below:

1 participant was rejected due to failing more than two out of the eight (two in each block) clearly telegraphed attention checks where participants were asked at two times in each block to press the arrow-key up before a timer of 10 seconds ran out. Four of these attention checks (one in each block) were shown directly following the last instructions before the start of the block and four (one in each block) were shown at randomly selected trials after the last trial of with a stimulus.

Results section

Model agnostic results

“S4. L. 146 grammatical issue? Rephrasing might be helpful here”

We thank the reviewer for catching this error. The sentence has now been reworded as follows: We anticipated that participants would use the values associated with the choices and their outcomes to guide their decisions.

“S5. For the Bayesian logistic regression, it would be interesting to report some measures of model convergence (trace plots, effective sample size (ESS), R-hat) in the supplements.”

We thank the reviewer for this suggestion. Supplementary plots 1 to 8 now show the trace plots and effective samples and R-hat is reported for each estimate. As can be seen there and below, all models converged nicely.

Robber block 1. Model 1: all Rhat = 1, except for correlation between the group-level subject’s effects of success and offer - cost (Rhat = 1.01); Model 2: all Rhat = 1.

Robber block 2. Model 1: all $R_{hat} = 1$, except for correlation between the group-level subject's effects of success and offer - cost ($R_{hat} = 1.02$); Model 2: all $R_{hat} = 1$, except for group-level effect for success probability ($R_{hat} = 1.01$).

Factory block 1. Model 1: all $R_{hat} = 1$, except for the correlations between group-level subject's effects of success last and offer-cost, and the correlations between group-level subject's effects of success last and the interaction effect ($R_{hat} = 1.01$); Model 2: all $R_{hat} = 1$, except for the group-level effects for trial, success probability and the interaction effect ($R_{hat} = 1.01$).

Factory block 2. Model 1: all Rhat = 1; Model 2: all Rhat = 1, except for the group-level effect of trial and the interaction effect (Rhat = 1.01).

“S6. How were the priors defined (I am assuming uniform/ Gaussian?) and how sensitive are the modelling results to different choices of priors (i.e., different choices of mean and SD)?”

We understand that the reviewer is asking about the priors of the Bayesian logistic regression models. These priors were chosen before data collection as part of the pre-registration, and are described in the methods section under the subsection Statistical Analyses:

“The models were fitted with weakly regularising Gaussian priors with mean $M = 0$ and varying standard deviations for the intercept ($SD = 1.5$), population ($SD = 1$) and group-level effects ($SD = 1$). The prior for the group-level effects correlation matrix was set with a Lewandowski-Kurowicka-Joe (LKJ) distribution with the parameter set at 3 thereby favouring somewhat weaker correlations between group-level effects”

As can be seen below, these priors are liberal when compared to the observed posterior effects (see example from the robber task block below) and do not put strong constraints on the potential observed effects, but do still shrink estimates towards 0 making it sceptical towards larger effects.

Furthermore, these priors were selected based on prior predictive simulations (Gelman et al., 2020).

“S7. Another thing to check would be the distribution of the residuals to ensure the results are trustworthy”

We thank the reviewer for this suggestion and appreciate the concern that the results of the regressions are not trustworthy. To our knowledge there are no specific distributional assumptions for the residuals for a logistic regression such as the ones for linear regressions. Furthermore, plotting the residuals for a Bayesian regression is not as straightforward for frequentist regression models. In Bayesian regression, each point estimate is represented by a distribution. While the full distributions can be plotted for each participant and trial, perhaps a better way of visualising discrepancies between the data and the model is using posterior predictive plots showing the deviation between the data and the model. Below are these plots for each of the models and blocks, where each participant’s total number of active choices are plotted (dots) against the model estimates of total active choices shown as distributions for each participant (the heatmaps). As is clear from these plots, the choice data for most participants are very well fitted by the model, indicating that the results are trustworthy.

“S8. Generally: I suggest adding some descriptive results about task performance as part of the model-agnostic results (mean/ range of wins or losses across participants, ...)- this may help to understand the special situation in Block 1 of the factory task?”

We thank the reviewer for this suggestion. The following has now been added to the model agnostic results section: In a follow-up post-hoc analysis, which was not part of the preregistered protocol, we explored potential differences between the tasks. Participants chose the active option more frequently in the robber task than the factory task in both block 1 (Robber: $M = 0.6611$, $SD = 0.1193$; Factory: $M = 0.6140$, $SD = 0.1104$; Wilcoxon signed-rank test: $V = 25961$, $p = 6.314e-11$) and block 2 (Robber: $M = 0.6605$, $SD = 0.1363$; Factory: $M = 0.6044$, $SD = 0.1104$; $V = 27815$, $p = 3.196e-16$). We also saw a small, non-significant, difference in the number of successes participants experienced when they chose the active option in the first block in the different tasks, whereby participants experienced more successes in the robber task than the factory task (Robber: $M = 0.5121$, $SD = 0.0773$; Factory: $M = 0.5018$, $SD = 0.0933$, $V = 20886$, $p = 0.2208$). This was not apparent in the second block (Robber: $M = 0.5120$, $SD = 0.0821$; Factory: $M = 0.5135$, $SD = 0.0799$, $V = 19241$, $p = 0.994$). We further did not see statistically significant differences in the mean number of successes between the two blocks of the robber task ($V = 19071$, $p = 0.8927$) or factory task ($V = 17356$, $p = 0.22$).

Figure 1

“S9. C+D: it took me quite a while to understand these plots. The Y-axis label is not descriptive enough – suggestion: value of choice or sth similar. Also, pls describe the meaning of the dots in the legend (I am assuming that these were the possible sizes of the offers) –”

We apologise for the unclear labelling of the Y-axis. This has now been changed in accordance with the suggestion provided (“value of choice”). We have also added the following sentence to the figure description to more clearly describe what the dots signal: c) Offers and costs in the tasks. The x-axis shows the offers for each task. In the robber task, the offer is the amount of the tip that the participants may earn at the end of the night before encountering the robber. The three blue lines show the values of the active choice (if successful) at the different cost regimes employed in the tasks. The red lines show the value of the passive choice. The dots show the specific values present in the task.

“S9.1 Question: why did you decide to have many low offers in the robber task and more high offers in the factory task instead of distributing it uniformly? Was this a decision made based on pilot data? Would be good to explain this choice.”

We thank the reviewer for this question. As can be seen in the figure, including many low offers in the robber task and high values in the factory task means that more values are closer to the equality point between the value of the active and passive options. Note that, since the probability of success is not included in these values, the figures imply an expected probability of 100 % success when the active option is chosen. As such, with more pessimistic expected probabilities more of the active options values will be below the passive options value enabling a spread in active and passive decisions. This spread thereby enables higher recovery of the parameters of the prior with fewer trials than would be needed for a uniform spread. This has now been emphasised in the following section:

Crucially, this cost was related to the size of the tip by constructing the cost as a sum of a percentage of the offer and a set cost. In the robber task, the set costs were 10, 40 or 75 points and the concomitant percentages of the offers were 15 %, 5 % and 0 %. This resulted in 60 unique offer-cost combinations with the majority of these combinations resulting in small differences between the values of the active and passive options (see Fig. 1c).

And: The corresponding set and percentage costs for the factory task were 0, 50 or 100 points and 10 %, 5 % and 0 %, also resulting in small differences between the values of the two options for the majority of combinations (see Fig. 1d).

“S10. E+F: These plots would benefit from a legend explaining the meaning of the different colors. I understand that these are shown in G but I think this is a bit too indirect. I would recommend a heatmap-type legend here, with the colors ranging from blue to red and representing the different success probabilities from 7 to 92%

“S10. E+F: These plots would benefit from a legend explaining the meaning of the different colors. I understand that these are shown in G but I think this is a bit too indirect. I would recommend a heatmap-type legend here, with the colors ranging from blue to red and representing the different success probabilities from 7 to 92%

S11. I do not quite understand the meaning of the opacity of the line. Could you rephrase the sentence in the legend (L. 175, is quite a convoluted sentence), and add a remark about this in the figure itself as well? My goal with this comment is to improve the first-glance interpretability of the plots.”

We thank the reviewer for these comments and suggestions to improve the readability and interpretability of these plots. We have now added a heatmap-type legend in the top two plots. We have also switched the opacity of the lines to instead graph the percentages of participants that experienced each success probability in each successive trial. See the updated plot below. We have also rephrased the legend to clarify this: The line shown in the bottom two plots and the opacity of the lines indicates the percentage of participants who encountered each success probability on each trial, as the number of times each participant encountered each of the success probabilities ranged from 4 (100 %) to 8 (11 %) trials. The lines thereby signal the increased uncertainty in the estimates, since not all participants encountered all stimuli 8 times.

To conform with the formatting guidelines, beyond these additions, we have now changed the range of the y-axis from 0.3-0.9 to 0-1. We hope these changes have improved the readability and interpretability of the plots. See the full figure below:

Bayesian models

“S12. Four different choice models were developed. They are described in more detail in the supplements, but it would be helpful to have a tabular (or graphical) overview of the models also in the main text, especially displaying what their choice function looks like and how they differ from one another in mathematical terms.”

We apologize for the unclear description of the models and hope that the revised version of the manuscript is clearer. First, now the models are fully described in the Methods section in detail. To improve readability and more clearly distinguish the models, we have, in accordance with your suggestion, included the following table in the text of the Methods section:

Name	N	Utility function	Choice function
SoftMax	1	Robber: $\begin{cases} Q_t(\text{go}) = o_t - c_t \\ Q_t(\text{ng}) = 0 \end{cases}$ Factory: $\begin{cases} Q_t(\text{go}) = -c_t \\ Q_t(\text{ng}) = -o_t \end{cases}$	$P_t(\text{go}) = \frac{1}{1 + e^{-(Q_t(\text{go}) - Q_t(\text{ng}))/\tau}}$
Win Stay Lose Shift	3	Robber: $\begin{cases} Q_t(\text{go}) = (o_t - c_t) + (\text{success}_{t-1} \cdot \gamma^+) - (\text{loss}_{t-1} \cdot \gamma^-) \\ Q_t(\text{ng}) = 0 \end{cases}$ Factory: $\begin{cases} Q_t(\text{go}) = -c_t + (\text{success}_{t-1} \cdot \gamma^+) - (\text{loss}_{t-1} \cdot \gamma^-) \\ Q_t(\text{ng}) = -o_t \end{cases}$	
Static Probability	2	Robber: $\begin{cases} Q_t(\text{go}) = -c_t + p \cdot o_t \\ Q_t(\text{ng}) = 0 \end{cases}$ Factory: $\begin{cases} Q_t(\text{go}) = -c_t - (1 - p) \cdot o_t \\ Q_t(\text{ng}) = -o_t \end{cases}$	
Bayesian Learner	3	Robber: $\begin{cases} Q_{j,t}(\text{go}) = -c_{j,t} + \mu_{0+t,j} \cdot o_{j,t} \\ Q_{j,t}(\text{ng}) = 0 \end{cases}$ Factory: $\begin{cases} Q_{j,t}(\text{go}) = -c_{j,t} - (1 - \mu_{0+t,j}) \cdot o_{j,t} \\ Q_{j,t}(\text{ng}) = -o_{j,t} \end{cases}$	

“S13. I am assuming that you used uninformative priors for all models, correct? How sensitive are the modelling results when choosing different parameters for the priors?”

We thank the reviewer for these questions. As stated in the Methods section, the prior distributions for all models and parameters were zero-centered gaussian distributions with the variance set at 6.25: “Each parameter in all models had a prior that was a zero-centred normal distribution with a variance of 6.25, as recommended by the authors of the toolbox”.

These priors were chosen based on the recommendation of the authors of the toolbox to ensure unbiased estimation of the parameters as discussed in detail in the supplementary material by Piray et al., (Piray et al., 2019). While we appreciate the usefulness of testing these assumptions and making sure the results are robust, without a suggestion of specific alternative priors to test we find it difficult to make a selection of other priors that would be better suited for testing our hypothesis and estimating the parameters or testing the robustness of our results.

“S13.1. (Reproducibility of the results is a bit hard to judge bc access to code is not granted yet)”

As requested, full access to code is now openly available for the reviewers at <https://github.com/Granwald/healthyPriors> . The anonymized data is openly available at doi:10.5281/zenodo.13745011.

“S14. Can you make the motivation/ conceptual thinking for the model augmentations, i.e., the action bias a bit clearer? L. 275”

We thank the reviewer for this comment, and apologize for being unclear in our motivations for testing model augmentations. The augmentations were tested as an attempt to remove noise from the parameters of interest or to test alternative hypotheses about the extent of learning. This description is now added to the Methods section: *These augmentations are tested as an attempt to remove noise from the estimated parameters of interest and to test alternative hypotheses about the extent of learning.*

For the bias parameter, we did not have any specific hypothesis but, when they survive model comparison, they improve our estimate of the prior. This has now been described in detail in the Methods: This implementation thereby assumes that there is a probability that the participant completely neglects the value calculation in preference of choosing a specific option. When surviving model comparison, this component prevents such decisions from affecting the estimate of the prior or other model parameters, much in the same way as implementations of lapse parameters in psychometric functions can improve the fit of these functions (Wichmann & Hill, 2001).

The continuous learning augmentation of the model assumes that participants learned the success probability across stimuli, that is they do not revert to the prior with each new stimulus. Thus, this model challenges the assumption that participants revert back to their subjective prior with each new stimulus. Further, emphasis has now been added to the Methods section: One augmentation was a model that did not revert to the prior with each new stimulus but instead continued to update the posterior from the last trial regardless of stimulus identity. This challenges the assumption of only updating within a stimulus.

“S15. L. 342, legend of Fig. 2: what does design matrix refer to here? Please specify”

We apologize for not being as detailed as needed. We have now added the following to the legend of the figure: (the specific order of offers, costs, success probabilities and number of trials with a stimulus)

“S16. Fig. 4: I think Block 1 and 2 in the follow up experiment refer to the different tasks, correct? Or do the plots display prior parameters across tasks? Please specify in title or so from what task(s) the data come.”

We thank the reviewer for this question and suggestion. The figures show the test-retest reliability of the task-invariant prior fitted across tasks. Block 1 and Block 2 here refers to the first and second block of both tasks. The title of the figure has now been changed to: Test-retest reliability for the prior fitted across tasks.

“S17. The model comparison is performed at the group-level using protected-exceedance probabilities. This allows some conclusions about which model is the most likely at the population level. This approach does not balance between goodness of fit and model complexity, like classical information criteria. I recommend evaluating the models with a measure that takes model complexity into account.”

We apologise for this confusion. While protected-exceedance probabilities, in their calculation, do not directly implement a balance between goodness-of-fit and model complexity, model frequency, which is used to calculate protected-exceedance probability, does. This has been clarified in the section on inference criteria: M.freq is an estimate of the frequency of participants that is best explained by the model (taking relative complexity into account). This is the estimate which was used to calculate the protected-exceedance probability.

How this balance is implemented in the probability that each individual subject is best fitted by a specific model is described in the method section (now with added emphasis): In model comparison, HBI applies a form of what is known as an automatic Occam factor in which more complex models are directly penalised by spreading probability mass more broadly, thereby balancing model complexity and goodness-of-fit. This stems from the extra level of the Bayesian hierarchy that it includes compared with the type-2 maximum likelihood iBIC method employed by Huys et al.

“S18. Have you considered reporting an absolute measure of goodness of fit, at least for winning models – akin to e.g. R^2 ?”

We thank the reviewer for this suggestion and apologize for this oversight in our previous reporting. We now report pseudo- R^2 for the winning models in each task and block separately: Results showed that the choice data in the robber task was best fitted by the unaugmented Bayesian learner model (Block 1: p - r^2 25th percentile = 0.4282, median = 0.5782, 75th percentile = 0.7002; Block 2: 25th percentile = 0.4372, median = 0.5728, 75th percentile = 0.7349) whereas the choice data in the factory task was best fitted by a model that included an action bias on top of the Static probability model in block 1 (p - r^2 25th percentile = 0.4429, median = 0.6453, 75th percentile = 0.7837) and the basic Bayesian model in block 2 (p - r^2 25th percentile = 0.4958, median = 0.6910, 75th percentile = 0.8171); for full results see Supplementary results).

and for the final 1-prior model in both blocks on the first day: As shown in Fig. 3, this model captures the observed behaviour well (p - r^2 for Block 1 at 25th percentile = 0.4409, median = 0.5832, and 75th percentile = 0.7083; Block 2: 25th percentile = 0.4581, median = 0.6010, and 75th percentile = 0.7409).

And at follow-up: The follow-up session consisted of one block of each task with new stimuli, and choice data were modelled using the one prior model (p - r^2 at 25th percentile = 0.5394, median = 0.6437, and 75th percentile = 0.7623).

The following description of this measure has also been added to the Methods section: For the winning models, we also report pseudo r^2 (p - r^2), a measure of the degree to which the data are explained by the model, normalized for the data likelihood under chance (Daw, 2011). This is calculated as the 1 - (log-likelihood of the model for a participant/log-likelihood under chance), meaning that numbers below 0 indicate worse fit than chance and 1 indicate a perfect fit. This measure is reported for the 25th, 50th and 75th percentile of participants.

Discussion

“L. 414-425, I find the argumentation here very convincing and agree that the authors, with their design have demonstrated an interesting way of studying Bayesian computational markers in behavioral experiments!”

We thank the reviewer for this positive comment!

Limitations

“S19. A clear limitation to the interpretations of a robust prior for avoidance that requires addressing is the less robust behavioral effect in the factory task, with many hypothesized effects showing up only in the second block – the reasons for this would need to be discussed (slower learning but why?) and I would appreciate some ideas on what happened in Block 1, whether these tasks maybe differed in other, unrelated aspects: difficulty? Motivation? The best fitting model seemed to be a static model with an overall action bias for Block 1 in the factory task.”

We thank the reviewer for pointing this out. While we cannot fully know why this apparent lack of updating occurs in the first block, we now offer one possible reason in the Discussion: One possible reason for this apparent lack of updating in the factory task is that participants chose the active action to a lesser extent in this block of the task compared to the robber task and the second block. This thereby provided fewer opportunities to sample participants' potential learning in this block. This difference in the tendency to choose the active

action may stem from the fact that the factory task lies in the negative domain in which loss aversion predominates (Kahneman & Tversky, 1979) in a way that might potentially initially increase the difficulty of processing and calculating the value difference.

“S20. The correlation between positive affect and prior mu is relatively tender (Fig. 5a&b) – interpret with care esp. with respect to the implications for affective disorders”

We thank the reviewer for this suggestion and have now significantly scaled back our conclusions and the implications for affective disorders. The section on the association between the mean of the prior and positive affect now reads:

While the correlation between positive affect and the mean of the prior adds some credence to the link between the prior and depressive symptoms, the correlation was rather weak and the association fails to cross our threshold for statistical significance when participants who failed attention checks were removed. Positive affect was hypothesised to be associated with the prior, since reduced positive affect is central in depression^{34,35,50,51} and a promising target for treatment^{50,52,53}. Furthermore, positive affect has large variance in the general population³⁶, which we therefore used as a proxy for depressive symptoms in a non-clinical sample. This weak association may indicate that the mean of the prior is not strongly related to depression symptoms. However, it may also mean that positive affect is an imprecise measure of anhedonia, which has itself been suggested to be closely related to expectation of controllability of the environment⁵⁵. We hypothesise that other measures of anhedonia would be more closely linked to the measured prior. However, the variance of anhedonia in a non-clinical sample such as the one we had is expected to be low compared with clinical populations⁵⁶. Future studies should examine the relationship between our prior and anhedonia in a clinical sample of depressed patients.

Minor language issues

“- L. 208: played with the low probability stimuli – suggest to say something like: encountered the stimuli with low success rates, to keep it consistent throughout.

- L. 236 too much italics

- L. 414: Replace “That” with “The fact that””

We thank the reviewer for their thorough comments! These language issues have now been addressed.

Reviewer #2:

“Thank you for the opportunity to review this manuscript. The authors have leveraged an innovative experimental design to determine the extent to which prior beliefs about the outcomes of actions are generalisable across different contexts. Although previous researchers have examined the influence of prior beliefs on decision making, these have been restricted to single task designs that are unable to determine whether the influence of individual differences in prior beliefs generalises across different tasks. By using two distinctive, aversively framed tasks to examine the influence of prior beliefs about the probability of success when performing an action to avoid potentially negative outcomes, the authors convincingly show that participants' prior beliefs reliably influenced decision making across different contexts. Importantly, by providing evidence of the convergent validity of the measured prior by demonstrating its correlation with self-report measures of positive affect, the authors results point to a Bayesian conceptualisation of helplessness, which has clear implications for our understanding of depression - namely, their results suggest that depressed states may be characterised by a generalisable pessimistic prior favouring active avoidance across different contexts.

In my view, the authors' submission is convincing, innovative and elegant, and fills an important gap in the literature. I am happy to recommend it for publication, in its current form, and I wish them all the best for their future endeavours.”

We thank the reviewer for these positive comments!

References added and part of responses:

Daw, N. D. (2011). Trial-by-trial data analysis using computational models: (Tutorial Review). In

M. R. Delgado, E. A. Phelps, & T. W. Robbins (Eds.), *Decision Making, Affect, and Learning: Attention and Performance XXIII* (p. 0). Oxford University Press.

<https://doi.org/10.1093/acprof:oso/9780199600434.003.0001>

Gelman, A., Vehtari, A., Simpson, D., Margossian, C. C., Carpenter, B., Yao, Y., Kennedy, L.,

Gabry, J., Bürkner, P.-C., & Modrák, M. (2020). *Bayesian Workflow* (arXiv:2011.01808).

arXiv. <https://doi.org/10.48550/arXiv.2011.01808>

Kahneman, D., & Tversky, A. (1979). Prospect Theory: An Analysis of Decision under Risk.

Econometrica, 47(2), 263. <https://doi.org/10.2307/1914185>

Piray, P., Dezfouli, A., Heskes, T., Frank, M. J., & Daw, N. D. (2019). Hierarchical Bayesian

inference for concurrent model fitting and comparison for group studies. *PLOS*

Computational Biology, 15(6), e1007043. <https://doi.org/10.1371/journal.pcbi.1007043>

Wichmann, F. A., & Hill, N. J. (2001). The psychometric function: I. Fitting, sampling, and goodness of fit. *Perception & Psychophysics*, 63(8), 1293–1313.

<https://doi.org/10.3758/BF03194544>

Reviewer #1 (Remarks to the Author):

“Thank you for the opportunity to review a revised version of the manuscript on Bayesian Priors and Avoidance for Comms Psych. The authors have meticulously revised their work based on my previous comments. Their dedication makes me feel fortunate to be a part of such a diligent and committed crowd. The updates include a more concise introduction, more thorough descriptions of the computational models as well as additional measures of fit, updated legends and revised figures with improved readability. “

We thank the reviewer for their kind comments, and thank the reviewer once again for their thorough reading and comments on the previous version of the manuscript.

“After re-analyzing the dataset without 3 participants who failed the attention checks, the correlation between the prior μ and positive affect was no longer significant. I am unsure how convincing I find the argument that this is due to a loss of power. However, since the authors now also discuss the implications of this correlation much more carefully in the discussion section, I think that this is acceptable. I suggest updating the sentence in the abstract about this correlation (manuscript p. 1 l. 28-29), since many readers will probably only read the abstract and should get more context about this finding.

Other than that, I am happy to recommend this interesting and well-executed manuscript for publication.”

We thank the reviewer for this suggestion and have now reworded the abstract to emphasize the precarious nature of this correlation:

The parameters of this prior were reliable, and participants' self-rated positive affect was weakly correlated with this prior such that participants with an optimistic prior reported higher levels of positive affect.

Reviewer #2 (Remarks to the Author):

Although my own review required no further modifications to the manuscript, it is clear from the authors' extensive responses to the first reviewer that they have taken this feedback seriously and gone to considerable lengths to address any concerns about the suitability of this manuscript for publication. Once again, I applaud the authors' efforts, recognise the unique and meaningful contribution of their research to the literature, and am happy to recommend its publication in Communications Psychology.

We thank the reviewer for these kind comments!